# Radial somatic F-actin organization affects growth cone dynamics during early neuronal development

Durga Praveen Meka[1],[†],[*] (ID), Robin Scharrenberg[1],[†] (ID), Bing Zhao[1], Oliver Kobler[2] (ID), Theresa König[1], Irina Schaefer[1], Birgit Schwanke[1], Sergei Klykov[3], Melanie Richter[1], Dennis Eggert[4],[5], Sabine Windhorst[6], Carlos G Dotti[7] (ID), Michael R Kreutz[8],[9] (ID), Marina Mikhaylova[3] (ID) & Froylan Calderon de Anda[1],[**],[‡] (ID)

## Abstract

The centrosome is thought to be the major neuronal microtubule-organizing center (MTOC) in early neuronal development, producing microtubules with a radial organization. In addition, albeit *in vitro*, recent work showed that isolated centrosomes could serve as an actin-organizing center, raising the possibility that neuronal development may, in addition, require a centrosome-based actin radial organization. Here, we report, using super-resolution microscopy and live-cell imaging of cultured rodent neurons, F-actin organization around the centrosome with dynamic F-actin aster-like structures with F-actin fibers extending and retracting actively. Photoactivation/photoconversion experiments and molecular manipulations of F-actin stability reveal a robust flux of somatic F-actin toward the cell periphery. Finally, we show that somatic F-actin intermingles with centrosomal PCM-1 (pericentriolar material 1 protein) satellites. Knockdown of PCM-1 and disruption of centrosomal activity not only affect F-actin dynamics near the centrosome but also in distal growth cones. Collectively, the data show a radial F-actin organization during early neuronal development, which might be a cellular mechanism for providing peripheral regions with a fast and continuous source of actin polymers, hence sustaining initial neuronal development.

**Keywords** actin; centrosome; microtubules; neuronal development; PCM-1
**Subject Categories** Cell Adhesion, Polarity & Cytoskeleton; Neuroscience

## Introduction

The centrosome is thought to be the major neuronal microtubule-organizing center (MTOC) in early developing neurons [1–3], producing microtubules with a radial organization [4,5]. Recently, it has been shown that isolated centrosomes can serve as an actin-organizing center *in vitro* [6], suggesting that the centrosome might control F-actin organization and dynamics during initial neuronal development. However, initial attempts to demonstrate that somatic F-actin can be delivered rapidly to distal growth cones were not successful [7,8]. Moreover, the classical view on the role of actin on neuronal development is contrary to this idea. For instance, numerous studies have demonstrated that F-actin is assembled locally in growth cones and that impaired local assembly is sufficient to affect neurite growth [9–14]. Nevertheless, it has been reported that growth cone-like structures, comprised of F-actin, have an antero-grade wave-like propagation along neurites, supporting neurite extension [15–18]. Additionally, an anterograde F-actin flow was described during neuronal migration [19,20] and at the base of growth cones [21]. These studies add weight to the possibility that centrifugal actin forces starting in the cell body may contribute to the neuronal polarization during development. To test this possibility, we performed a series of state-of-the-art experiments to examine somatic F-actin organization and dynamics in living neurons.

Mechanistically, we propose PCM-1 as a molecular determinant for somatic F-actin organization. PCM-1 has been shown to promote F-actin polymerization in non-neuronal cells [6], and PCM-1-containing pericentriolar satellites are important for the recruitment of proteins that regulate centrosome function [22]. The depletion of PCM-1

1  RG Neuronal Development, Center for Molecular Neurobiology Hamburg (ZMNH), University Medical Center Hamburg-Eppendorf, Hamburg, Germany
2  Combinatorial Neuroimaging Core Facility (CNI), Leibniz Institute for Neurobiology, Magdeburg, Germany
3  Emmy-Noether Group "Neuronal Protein Transport", Center for Molecular Neurobiology (ZMNH), University Medical Center Hamburg-Eppendorf, Hamburg, Germany
4  Max Planck Institute for the Structure and Dynamics of Matter, Hamburg, Germany
5  Heinrich Pette Institute—Leibniz Institute for Experimental Virology, Hamburg, Germany
6  Department of Biochemistry and Signal Transduction, University Medical Center Hamburg-Eppendorf, Hamburg, Germany
7  Centro de Biología Molecular "Severo Ochoa", CSIC-UAM, Madrid, Spain
8  RG Neuroplasticity, Leibniz Institute for Neurobiology, Magdeburg, Germany
9  Leibniz Guest Group "Dendritic Organelles and Synaptic Function", Center for Molecular Neurobiology (ZMNH), University Medical Center Hamburg-Eppendorf, Hamburg, Germany
   *Corresponding author. Tel: +49 40 7410 56877; E-mail: praveen.meka@zmnh.uni-hamburg.de
   **Corresponding author. Tel: +49 40 7410 56817; Fax: +49 40 7410 56450; E-mail: froylan.calderon@zmnh.uni-hamburg.de
   †These authors contributed equally to this work
   ‡Lead contact

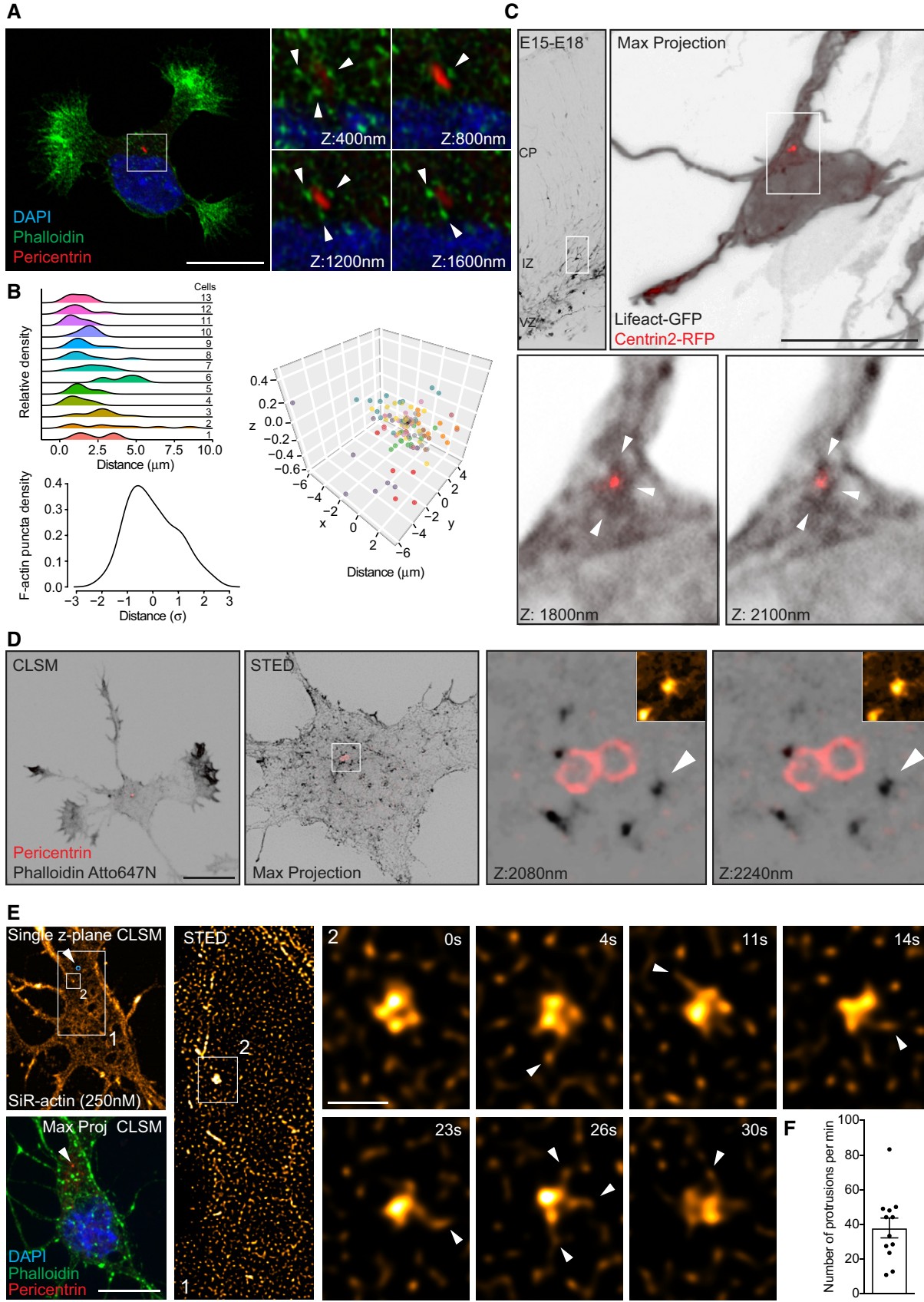

**Figure 1.**

**Figure 1. Super-resolution microscopy reveals cytosolic F-actin puncta releasing F-actin fibers in developing neurons.**

A   Stage 2 hippocampal neuron labeled with phalloidin and Pericentrin antibody, confocal z-stacks from the inset show F-actin puncta around the centrosome. Arrowheads point F-actin surrounding the centrosome.
B   Cytosolic F-actin puncta tend to be closer to the centrosome. Ridge plots indicate the density of puncta for individual cells in dependence of distance to the centrosome. Normalization of the distance in each cell reveals a skewed distribution toward the centrosome. Values are centered around the mean and expressed as standard deviations from the mean (z-score). As an overview, puncta coordinates (color-coded for individual cells) are shown in 3D with cells aligned at the centrosome (black spot).
C   Multipolar cell located in the IZ of the developing cortex expresses Lifeact-GFP and Centrin2-RFP and shows F-actin puncta surrounding the centrosome in the developing cortex. Arrowheads point F-actin surrounding the centrosome.
D   Confocal (CLSM) and STED microscopic images of stage 2 hippocampal neuron. Inset: STED Z-stack images with 160 nm Z-spacing showing F-actin puncta localizing near the centrosome. Insets from arrowheads in Z-stack images show F-actin puncta with F-actin fibers attached.
E   Single Z-plane image (upper left panel) of stage 2 neuron labeled with SiR-actin (250 nM) for STED live imaging, confocal maximum projection image (lower, left panel) of the same cell which was later PFA-fixed and stained with Phalloidin, pericentrin, and DAPI. The position of the centrosome is depicted in both panels (white arrowheads). Inset 1: snap shot of STED time lapse. Time-lapse montages from inset two depict a single F-actin punctum releasing F-actin fibers (white arrowheads). Time (t) interval = 1.7 s (Movie EV1).
F   Quantification of protrusion frequency of F-actin fibers from the somatic F-actin puncta. The average extension frequency per min = $37.91 \pm 5.713$. Mean $\pm$ SEM; 12 puncta analyzed from six cells.

Data information: Scale bar: 10 μm (A, C and D); 5 μm (E); 0.51 μm (inset E).

disrupts the radial organization of microtubules without affecting microtubule nucleation [22]. PCM-1 particles preferentially localize near the centrosome [23,24]. In previous work, we found PCM-1 down-regulation in the developing cortex disrupted neuronal polarization, precluded axon formation, and impaired neuronal migration [24]. Here, we demonstrate that PCM-1 determines not only the somatic F-actin organization and dynamics but also that lack of PCM-1 has a radial effect, which ultimately disrupts growth cone dynamics and neurite length. Overall, our data show a novel somatic F-actin organization regulating early neuronal development *in vitro* and *in vivo*.

## Results

### F-actin organizes around the centrosome

Given the lack of knowledge regarding somatic F-actin organization, we sought to investigate the micro- and nano-structural organization of cytosolic F-actin near the centrosome. To this end, we used confocal and super-resolution microscopy during early neuronal differentiation *in vitro* (from stage 1 to early stage 3; [25]) and *in situ*. F-actin cytoskeleton in fixed and live cells was visualized via confocal and STED microscopy by labeling cells with Phalloidin-488, Phalloidin Atto647N, SiR-actin probe, or Lifeact-GFP [26]. Using confocal microscopy, we found a preferential localization of cytosolic F-actin puncta near the centrosome in cultured rat hippocampal neurons labeled with phalloidin and in neurons in the developing mouse cortex labeled with Lifeact-GFP (Fig 1A–C and Appendix Fig

S1A–C). STED microscopy images revealed that somatic F-actin organized as tightly packed structures constituted by a core of dense F-actin attached to F-actin fibers (aster-like structures, Fig 1D and Appendix Fig S1D). Moreover, we used single molecule localization microscopy (SMLM/STORM) to image Phalloidin-Alexa647 and corroborated that F-actin is organized around the centrosome in a pocket-like structure, where several F-actin puncta surrounded the centrosome with individual puncta exhibiting an aster-like organization (Appendix Fig S2).

In order to determine whether or not somatic F-actin puncta represent true sites of actin polymerization, we transfected cells with Lifeact-GFP and performed epifluorescence time-lapse imaging (frame rate: 2 s for 5 min) on DIV1 neurons. Cells expressing high levels of Lifeact showed stabilized F-actin in the form of somatic F-actin fibers (Appendix Fig S3A and D). Therefore, we exclusively analyzed cells where Lifeact expression labeled the F-actin cytoskeleton at similar levels as detected with phalloidin staining (Appendix Fig S3B–D). Time-lapse analysis of cells co-transfected with Lifeact-GFP and the microtubule plus-end marker EB3-mCherry corroborated that F-actin puncta concentrate near the MTOC identified by radial trajectories of EB3-mCherry comets (Fig EV1A and B). Moreover, our recordings showed that the F-actin puncta in the soma are highly dynamic and intermittent. These puncta exhibit a repetitive appearance and disappearance at the same location as shown via kymographs (Fig EV1C and D). Based on the duration of appearance, we categorized them as unstable (< 15 s), intermediately stable (16–240 s), and long-lasting (241–300 s) F-actin puncta. The majority of puncta are unstable (Fig EV1E), suggesting that these puncta are places of

**Figure 2. Somatic F-actin puncta act as rapid supply sources of F-actin to the periphery in developing neurons.**

A   PaGFP-UtrCH and tDimer co-transfected DIV1 rat hippocampal neuron photoactivated in the soma with 405 nm laser (red circle with a diameter of 5.239 μm).
B   PaGFP and tDimer co-transfected DIV1 rat hippocampal neuron photoactivated in the soma with 405 nm laser (red circle with a diameter of 5.146 μm).
C   Left panel: normalized intensity values in the photoactivated area of PaGFP-UtrCH and PaGFP expressing cells. Inset graph: half-time ($t\frac{1}{2}$) values in seconds for PaGFP-UtrCH cells = $15.71 \pm 0.712$ (n = 9), PaGFP cells = $16.06 \pm 1.755$ (n = 12). Middle panel: photoactivated signal in growth cones over time relative to the average initial signal from illuminated area for PaGFP-UtrCH and PaGFP expressing cells. Right panel: Growth cone to soma photoactivated signal intensity ratio of PaGFP-UtrCH and PaGFP expressing cells. All graphs: mean $\pm$ SEM; n = 9 cells for PaGFP-UtrCH, n = 12 cells for PaGFP from at least three different cultures.
D   Neurites from cells in (A; insets 1, 2) and (B; insets 3, 4) show the reach of the photoactivated signal at 94.4 s of the time lapse.
E   Normalized intensity values from the growth cones plotted against their neurite lengths of PaGFP-UtrCH and PaGFP expressing cells 120 s after photoactivation.

Data information: Scale bar: 10 μm.

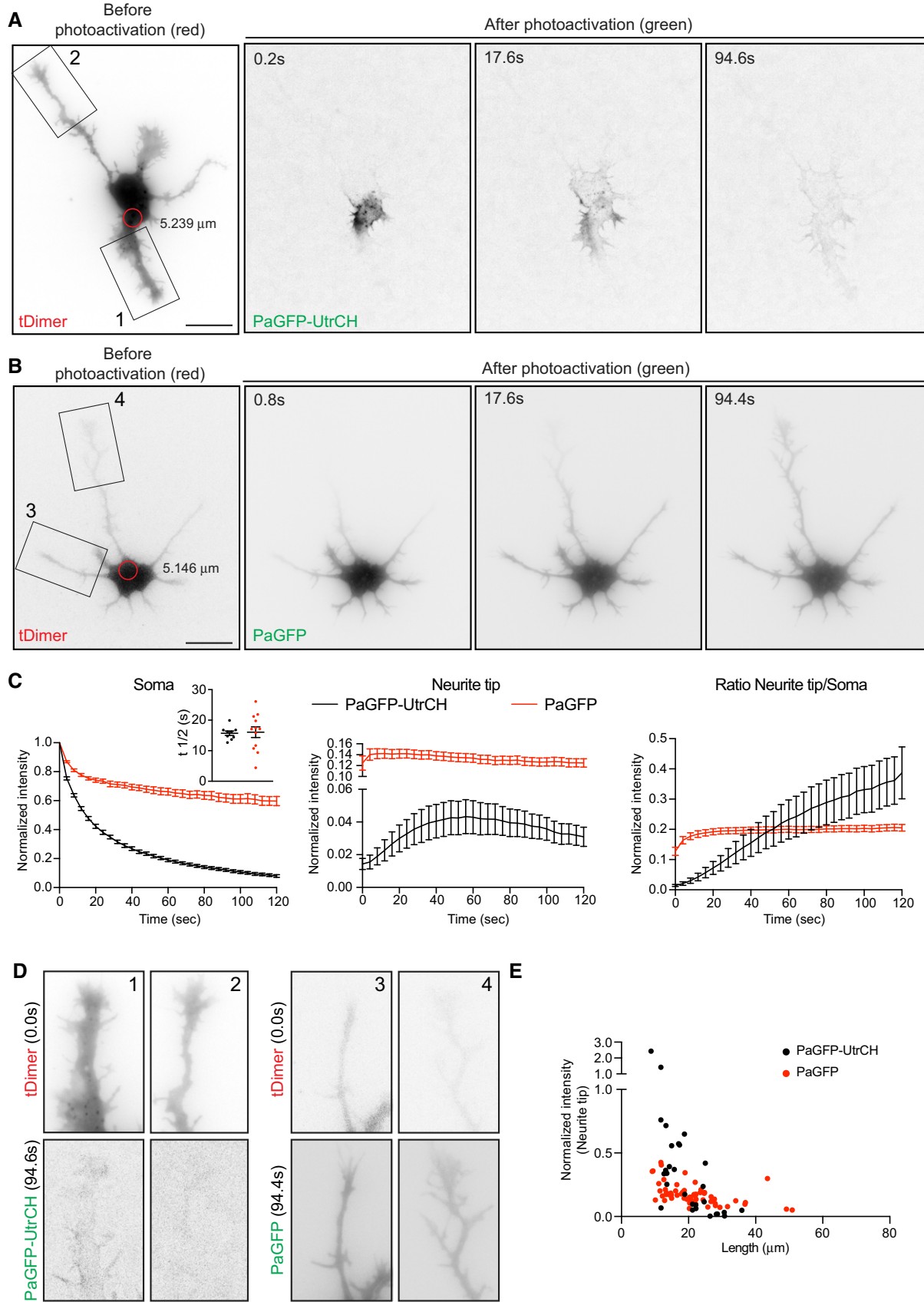

Figure 2.

high F-actin turnover. Accordingly, our FRAP analysis of somatic F-actin puncta showed fast fluorescence recovery (Appendix Fig S4A and B), thus confirming that somatic F-actin puncta are places of high F-actin turnover. We also found that these F-actin puncta in the cell body release F-actin comets (pointed by red arrowheads in Fig EV1D), which suggest that they might function as a source of somatic F-actin.

To gain further insight into the relevance of this F-actin organization, we employed STED time-lapse microscopy labeling F-actin with SiR-actin. Since SiR-actin is known to stabilize F-actin, we tested different concentrations of SiR-actin (250 and 500 nM). Cells labeled with 500 nM SiR-actin, with 5 h of incubation, showed less dynamic F-actin aster-like structures, with longer F-actin fibers attached to the F-actin core (Appendix Fig S5), whereas cells labeled with 250 nM SiR-actin, with 1.5–3 h of incubation, showed aster-like F-actin structures that are highly dynamic, extending F-actin fibers constantly in the range of seconds (Fig 1E and F; Movie EV1). Altogether, these results unveil the existence of a complex and dynamic somatic F-actin organization near the centrosome, suggesting a role in neuronal development.

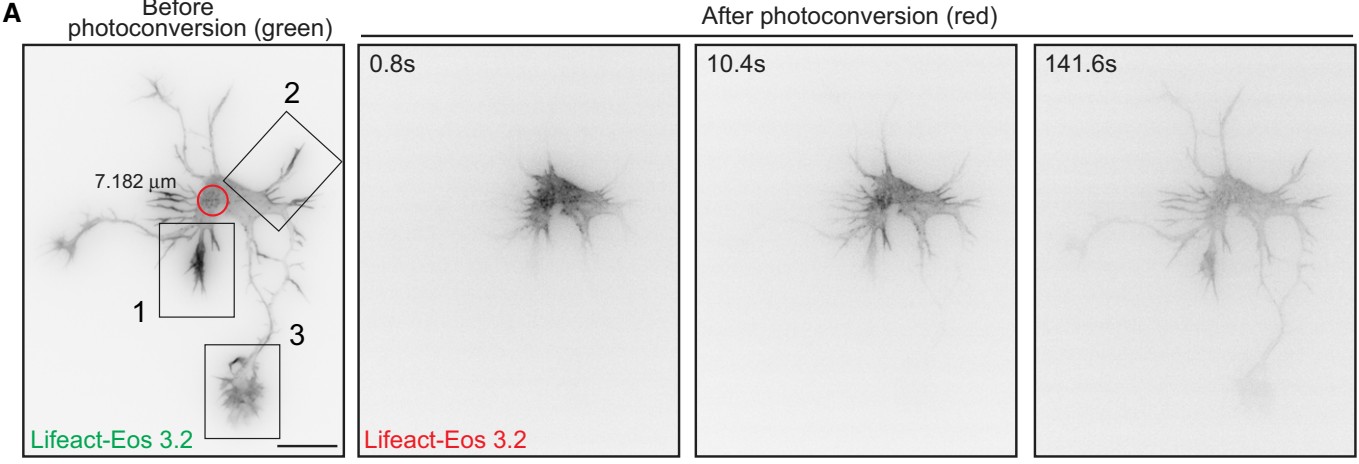

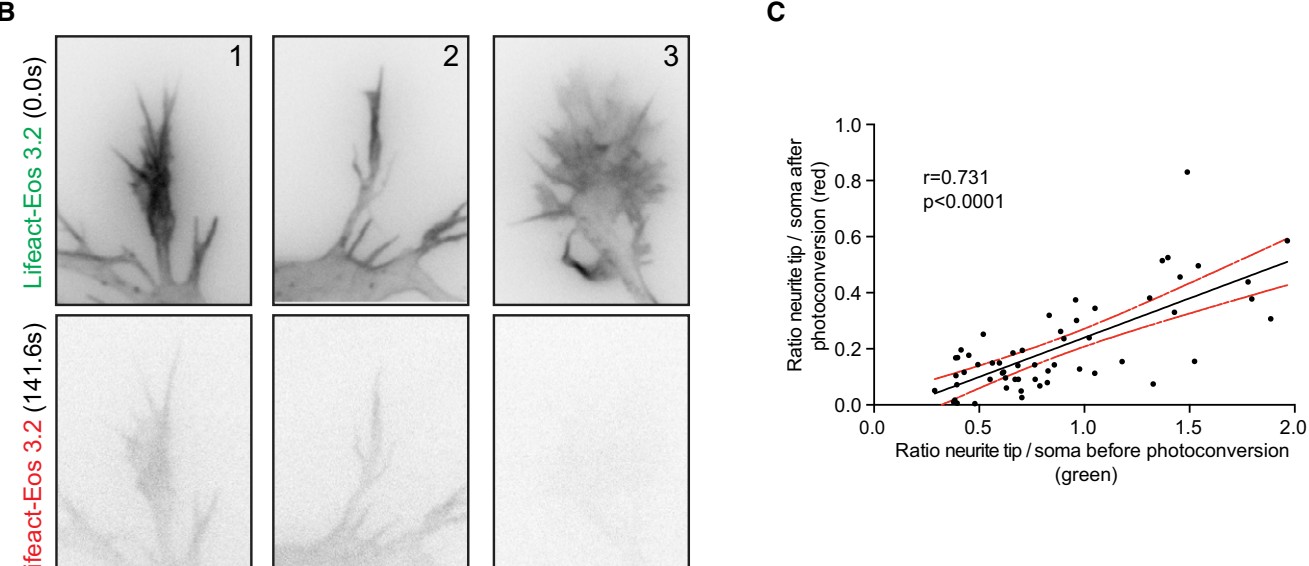

**Figure 3. Somatic F-actin translocation occurs preferentially to the growth cone with higher F-actin content.**

A   Cell expressing Lifeact-mEos3.2 showing preferential translocation of photoconverted Lifeact-mEos3.2 to the growth cone with higher content of Lifeact-mEos3.2 before photoconversion (inset 1).

B   Neurites from cell in (A; insets 1, 2, 3) show the reach of the photoconverted signal at the end of the time lapse (141 s).

C   Pearson correlation of growth cone to soma intensity ratio before and after photoconversion in Lifeact-mEos3.2 expressing cells ($n$ = 57 neurites; Pearson $r$ = 0.731, $P$ < 0.0001).

Data information: Scale bar: 10 μm.

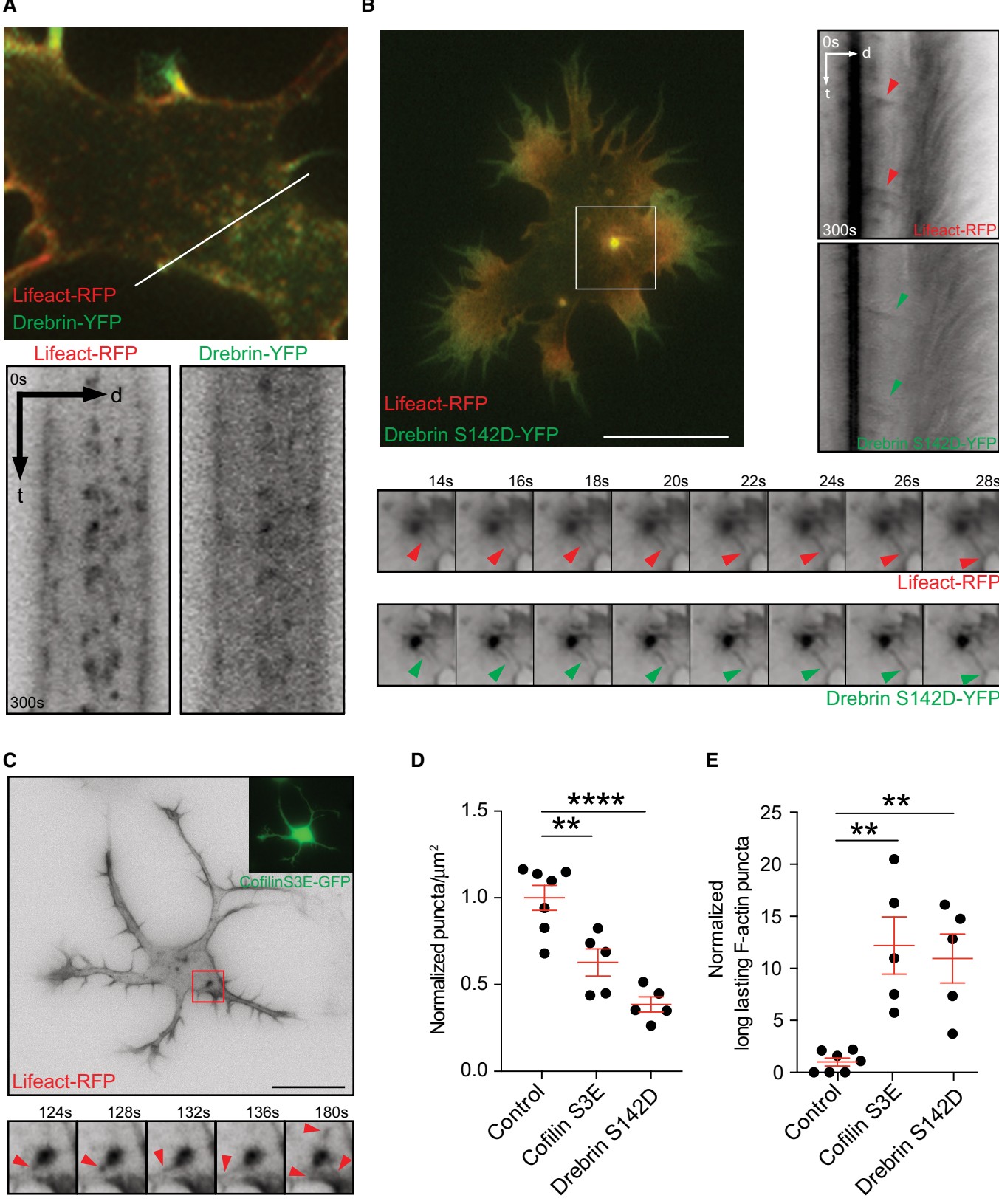

**Figure 4.**

◄

Figure 4. Stabilization of F-actin, by overexpressing Drebrin or Cofilin phosphomimetic mutant, unveils F-actin release from somatic puncta to the periphery.

A   Stage 2 cell transfected with Lifeact-RFP and Drebrin-YFP shows similar distribution of F-actin and Drebrin, highlighted via Kymograph.
B, C   Cells co-transfected with Lifeact-RFP and (B) Drebrin-S142D-YFP or (C) Cofilin-S3E-GFP stabilizes F-actin. Kymograph (in B) and time-lapse montages (in B and C), from insets, show F-actin asters releasing fibers to the cell periphery (arrow heads).
D   Density of somatic F-actin puncta of stage 2 cells in control group = 1.000 ± 0.07143, Cofilin-S3E group = 0.6277 ± 0.07836 and Drebrin-S142D group = 0.3856 ± 0.04340; $P < 0.0001$ by one-way ANOVA, *post hoc* Dunnett's test, **$P < 0.01$, ****$P < 0.0001$.
E   Comparison of long-lasting F-actin puncta in the soma of stage 2 cells control group = 1.000 ± 0.3787, Cofilin-S3E group = 12.20 ± 2.746, Drebrin-S142D group = 10.95 ± 2.346; $P = 0.0009$ by one-way ANOVA, *post hoc* Dunnett's test, **$P < 0.01$.

Data information: Mean ± SEM; $n = 7$ cells for control, $n = 5$ cells each for Cofilin-S3E and Drebrin-S142D groups from at least two different cultures (for data shown in D and E). Scale bar: 5 μm (A) and 10 μm (B, C).

## Somatic F-actin is radially delivered to the cell periphery

We therefore asked whether somatic actin polymerization could serve as a source for cell peripheral F-actin. For that aim, we used DIV 1 neurons transfected with photoactivatable GFP-Utrophin (PaGFP-UtrCH), which specifically labels F-actin [27,28], or photo-convertible Lifeact (Lifeact-mEos3.2). PaGFP-UtrCH photoactivation is irreversible in response to 405 nm light with an emission peak at 517 nm. Lifeact-mEos3.2 undergoes an irreversible photoconversion in response to 405 nm light from green to red fluorescence with emission peaks at 516–581 nm, respectively. We tested for the photobleaching of both probes and the initial fluorescence levels immediately after photoactivation/photoconversion to ensure that the values analyzed are similar within experiments (Appendix Fig S6A–J). We had to illuminate several F-actin puncta (5.2–7.1 μm diameter) at once given that single punctum illumination (2.2 μm diameter) did not yield enough traceable converted signal when spreading further (Appendix Fig S6K). Notably, we found that the somatic photoactivation of PaGFP-UtrCH leads to the distribution of photoactivated UtrCH signal to the cell periphery (Fig 2A, C and D; Movie EV2). Similarly, when we photoconverted a group of Lifeact-mEos3.2-labeled F-actin puncta in the soma, the intensity of the converted F-actin puncta decreased with time concomitant with a fast increase of converted signal in the neurite tips/growth cones (Fig EV2A, D and E). Another actin probe (actin-mEos4b), which labels F-actin and actin monomers, also distributed into growth cones after illumination (Fig EV2B, D and E). Further characterization of photoconverted Lifeact-mEos3.2 (red signal) or photoactivatable PaGFP-UtrCH in the cell periphery showed that translocation does not occur preferentially to the tips of the longest neurite (Figs 2E and EV2F) but to the neurite tips containing more F-actin (Fig 3A–C; Movie EV3). However, illumination of PaGFP or mEos3.2 alone resulted in distribution of the activated/converted signal independently of the neurite length (Figs 2B, D, E and EV2C, E, F, and Movie EV4).

To confirm that the radial translocation of UtrCH or Lifeact signal is due to the movement of F-actin but not actin monomers bound to UtrCH or Lifeact probes, we treated the cells with cytochalasin D, which disrupts the F-actin cytoskeleton. It has been shown that cytochalasin D treatment induces F-actin clusters around the centrosome in non-neuronal cells [6]. Similarly, we observed that cytochalasin D (1 μM) treatment in neurons induced the formation of F-actin aggregates near the centrosome from pre-existing intermittent F-actin puncta (Fig EV3A and B). These clusters do not depend on membrane organization since brefeldin A (BFA, 10 mg/ml)—which disrupts Golgi, endoplasmic reticulum, endosomes and lysosomes [29]—did not affect the localization of F-actin clusters near the

centrosome (Fig EV3C). Photoactivation/photoconversion of the somatic F-actin clusters of 1 μM cytochalasin D-treated cells did not induce UtrCH or Lifeact translocation toward the cell periphery (Fig EV3D–G, and Movie EV5), indicating that the translocation of photoactivated/photoconverted signal is not due to the movement of the Lifeact bound to actin monomers or freely diffusible UtrCH and Lifeact probes, but indeed labeled F-actin. Moreover, photoactivated PaGFP translocated to the cell periphery even after cytochalasin D treatment (Fig EV3D and E, and Movie EV5). Although Lifeact binds *in vitro* with higher affinity to actin monomers than to F-actin [30], it is still possible that in cells the amount of Lifeact bound to actin monomers is not the predominant species. This is also suggested by the pattern of Lifeact expression, which resembles the Phalloidin staining (Appendix Fig S3B–D). Altogether, our results suggest that filamentous actin translocates from the soma to the cell periphery.

Next, we tested whether or not F-actin translocation is exclusively radially oriented. Consequently, we decided to illuminate growth cones labeled with PaGFP-UtrCH, PaGFP, Lifeact-mEos3.2, actin-mEos4b, or mEos3.2. When PaGFP, mEos3.2, or actin-mEos4b transfected neurons were illuminated at growth cones, the converted signal translocated toward the cell body (Fig EV4B and C; Appendix Fig S7B, C and D; Movie EV6). In contrast, illumination of growth cones labeled with PaGFP-UtrCH or Lifeact-mEos3.2 did not induce retrograde movement of photoactivated UtrCH or photoconverted Lifeact signal to the cell body (Fig EV4A and C, and Appendix Fig S7A and D; Movie EV7), thus showing that F-actin translocation is unidirectional.

In order to decrease F-actin dynamics to better resolve the somatic F-actin translocation to the cell periphery, we treated cultured neurons with Jasplakinolide, an agent which stabilizes polymerized actin filaments and stimulates actin filament nucleation [31]. However, our treatment (0.3, 0.5, and 1 μM for 1.5–4 h) precluded the existence of peripheral F-actin and induced the formation of a somatic F-actin ring structure, detected both with Lifeact and with Phalloidin staining (Appendix Fig S8A and B). This structure appeared in the area with the highest density of plus-end microtubules labeled via EB3-mCherry (Appendix Fig S8B). These results suggest that Jasplakinolide prevents F-actin dynamics, thus "locking" the F-actin in the soma and blocking its movement. Photoactivation/photoconversion experiments further confirm these findings showing no radial movement of photoactivated UtrCH or photoconverted Lifeact signal to the periphery in 300 nM Jasplakinolide-treated cells (Appendix Fig S8C–F).

Importantly, with Drebrin or Cofilin constructs, as F-actin stabilizing tools, we could decrease the overall dynamics of F-actin without completely stabilizing/freezing the actin filaments. Drebrin inhibits Cofilin-induced severing of F-actin and stabilizes F-actin

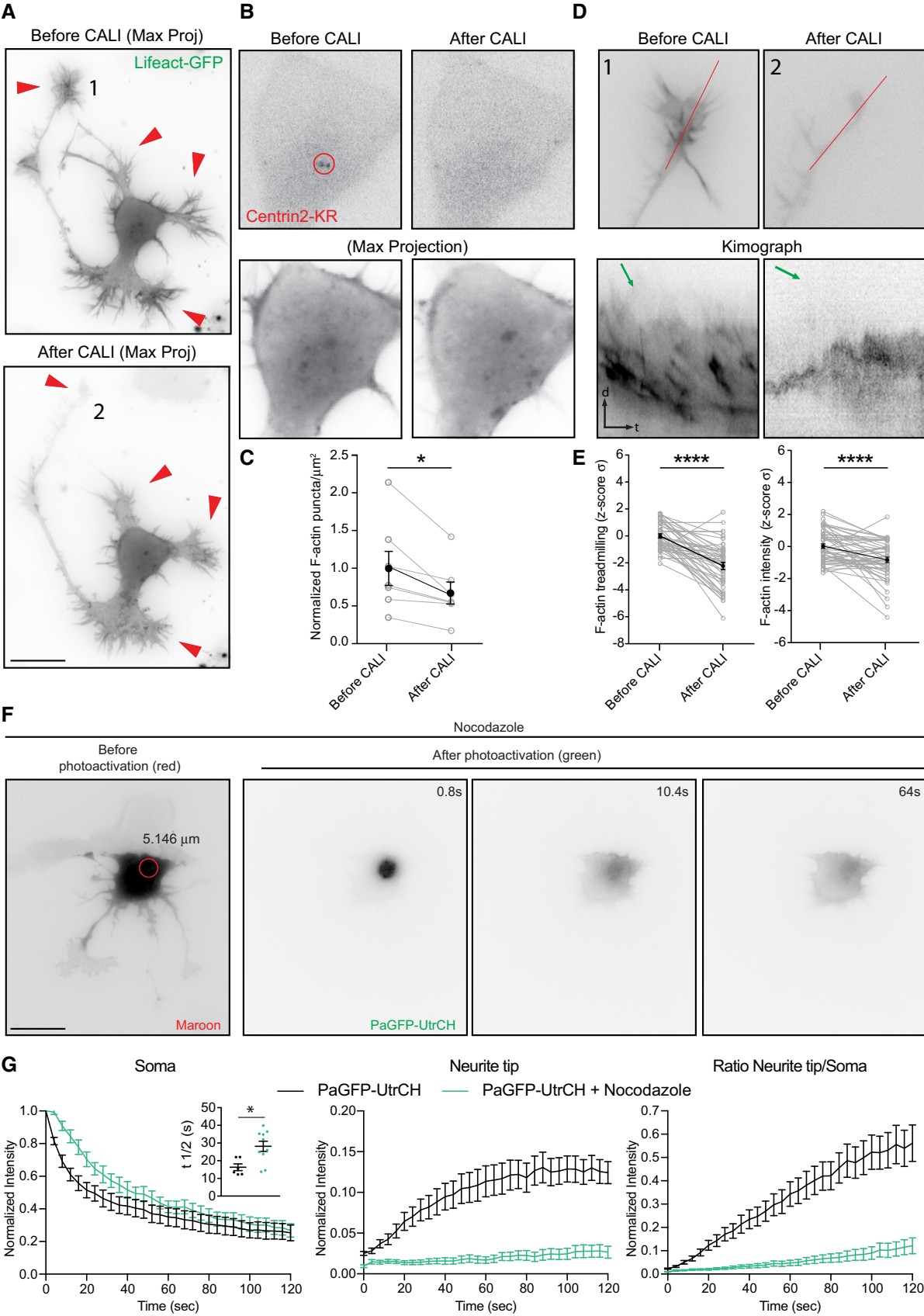

**Figure 5.**

◀

**Figure 5. Centrosome inactivation affects the number of somatic F-actin puncta and the F-actin intensity/F-actin treadmilling in growth cones.**

A   Neuron transfected with Centrin2-KillerRed and Lifeact-GFP subjected to localized CALI treatment. Red arrowheads show F-actin signal in growth cones before (upper panel) and after (lower panel) CALI of the centrosome.

B   Upper panel: Centrin2-KR signal before and after CALI. Lower panels: maximal projection of the time lapse of somatic F-actin signal.

C   CALI of centrosome decreases the number of F-actin puncta in the cells. Normalized F-actin puncta per $\mu m^2$ before CALI = 1.00 $\pm$ 0.226, in the same cells after CALI = 0.6737 $\pm$ 0.145. *P = 0.0205 by paired Student's *t*-test. Mean $\pm$ SEM; *n* = 7 cells from at least three different cultures.

D   Upper panels selected growth cones from cell in (A, 1 and 2) before and after CALI of the centrosome. Lower panel: Kymograph of actin treadmilling in growth cones of upper panels before and after CALI treatment. Green arrows on the kymographs indicate the slope of F-actin treadmilling in the neurite tips.

E   CALI of centrosome deceases F-actin treadmilling speed and F-actin intensity in growth cones. Values are centered around the mean and expressed as standard deviations from the mean (*z*-score). Before CALI = 0 $\pm$ 0.154, after CALI = −2.226 $\pm$ 0.265. Mean $\pm$ SEM; *n* = 10 cells from at least three different cultures. Paired *t*-test, ****P < 0.0001.

F   PaGFP-UtrCH and mMaroon1 co-transfected DIV1 rat hippocampal neuron treated with nocodazole (7 $\mu$M for 1.5 h) photoactivated in the soma with 405 nm laser (red circle with a diameter of 5.146 $\mu$m).

G   Left panel: normalized intensity values in the photoactivated area (soma) of untreated, nocodazole-treated PaGFP-UtrCH expressing cells. Inset graph: half-time (*t*½) values in seconds for untreated cells = 16.29 $\pm$ 1.884 (*n* = 6), nocodazole-treated cells = 28.14 $\pm$ 2.869 (*n* = 10). *P = 0.0106 by unpaired Student's *t*-test. Middle panel: photoactivated signal in the neurite tip over time relative to the average initial signal from illuminated area for untreated, nocodazole-treated PaGFP-UtrCH expressing cells. Right panel: neurite tip to soma intensity ratio of photoactivated untreated, nocodazole-treated PaGFP-UtrCH expressing cells. All panels: mean $\pm$ SEM; *n* = 6 for untreated cells, *n* = 10 for nocodazole-treated PaGFP-UtrCH cells, from at least two different cultures.

Data information: Scale bar: 10 $\mu$m.

[32–34]; Drebrin phosphorylation at S142 promotes F-actin bundling [35]. Therefore, the Drebrin phosphomimetic mutant (S142D) is a suitable candidate to decrease overall F-actin dynamics. Similarly, phosphomimetic Cofilin (S3E) is not able to sever F-actin; thus, it decreases F-actin turnover [36]. Previously, it was shown that Drebrin co-localizes with F-actin in growth cones [37]. Time-lapse microscopy analysis of Drebrin transfected cells revealed Drebrin to co-localize with F-actin puncta in the cell body (Fig 4A; Movie EV8). Importantly, the total number of somatic F-actin puncta decreased after Drebrin-S142D expression (Fig 4D) with an increase in the relative number of long-lasting F-actin puncta (Fig 4E). Interestingly, the stable F-actin puncta released noticeable F-actin comet-like structures toward the cell periphery (Fig 4B, Movie EV8).

Likewise, expression of Cofilin-S3E decreased total number of somatic F-actin puncta with an increment of long-lasting F-actin puncta, compared to cells expressing only Lifeact-RFP (Fig 4D and E). Furthermore, somatic F-actin puncta acquired an aster-like appearance releasing F-actin toward the cell cortex (Fig 4C). Finally, we analyzed the relevance of stable filaments on the radial movement of actin. We found that expression of PaGFP-UtrCH together with Cofilin-S3E leads to a slow movement of the activated signal in the cell body compared with control cells (Appendix Fig S9A–C). However, the intensity of the converted signal, which reaches the growth cone, is higher when the F-actin is stabilized (Appendix Fig S9A–C). Most likely this effect is due to decreased F-actin turnover in the growth cones caused by F-actin stabilization. Moreover, stabilization of F-actin with the expression of Cofilin-S3E leads to a reduced neurite length compared to cells expressing Cofilin-WT ([38], Appendix Fig S9D and E). Altogether, these results demonstrate that somatic F-actin puncta release F-actin toward the cell periphery, thus affecting early neuronal differentiation.

### Centrosomal activities affect F-actin in the cell periphery

Given that somatic F-actin puncta concentrate near the centrosome (Fig 1B), we asked whether centrosomal integrity is required for F-actin dynamics in developing neurons. We used chromophore-assisted light inactivation (CALI) based on the genetically encoded photosensitizer KillerRed, which upon green light illumination

(540–580 nm), will specifically inactivate the target protein via the generation of light-activated reactive oxygen species [39]. We fused Centrin2, a protein confined to the distal lumen of centrioles and present in the pericentriolar material, to KillerRed (Centrin2-KR) to specifically inactivate the centrosome with laser illumination (561 nm). Cells expressing Centrin2-KR and either EB3-GFP or Lifeact-GFP were imaged before laser illumination. We then locally illuminated the centrosome with the 561 nm laser for 1.5 s to inactivate Centrin2 specifically at the centrosomal region without affecting somatic Centrin2 (Fig 5A and B). Two to three hours after laser illumination, cells were reimaged. Centrosome inactivation via CALI reduced the number of somatic microtubules (Fig EV5A and B; Movie EV9) and somatic F-actin puncta (Fig 5B and C). Importantly, F-actin treadmilling speed as well as the F-actin intensity at the cell periphery was significantly reduced after centrosomal disruption (Fig 5D and E; Movie EV10). As a control, we illuminated a similar sized area at the soma away from the centrosome with the same settings. The cells illuminated outside the centrosomal area did neither show reduced F-actin treadmilling speed nor decreased F-actin intensity in growth cones (Fig EV5C and D; Movie EV11).

Microtubule organization in early developing neurons is centrosome-dependent [1–3]. Therefore, we decided to disrupt microtubule polymerization with nocodazole to test whether the F-actin translocation toward the cell periphery is affected. We found that 7 $\mu$M nocodazole treatment drastically reduced the motility of photoactivated (or photoconverted) UtrCH (or Lifeact) signal from the soma to the periphery (Figs 5F and G, and EV5E and F). Accordingly, microtubules disruption in developing neurons leads to a less dynamic F-actin cytoskeleton [34]. Altogether, these results show that the centrosome and microtubules are necessary for somatic F-actin translocation toward the cell periphery.

### PCM-1 determines somatic F-actin organization

Next, we tested PCM-1 as a molecular determinant of F-actin dynamics near the centrosome. PCM-1 promotes F-actin polymerization in non-neuronal cells [6]. We found that PCM-1 particles intermingled with F-actin puncta in the soma of fixed neurons and concentrated in proximity of F-actin puncta (average distance

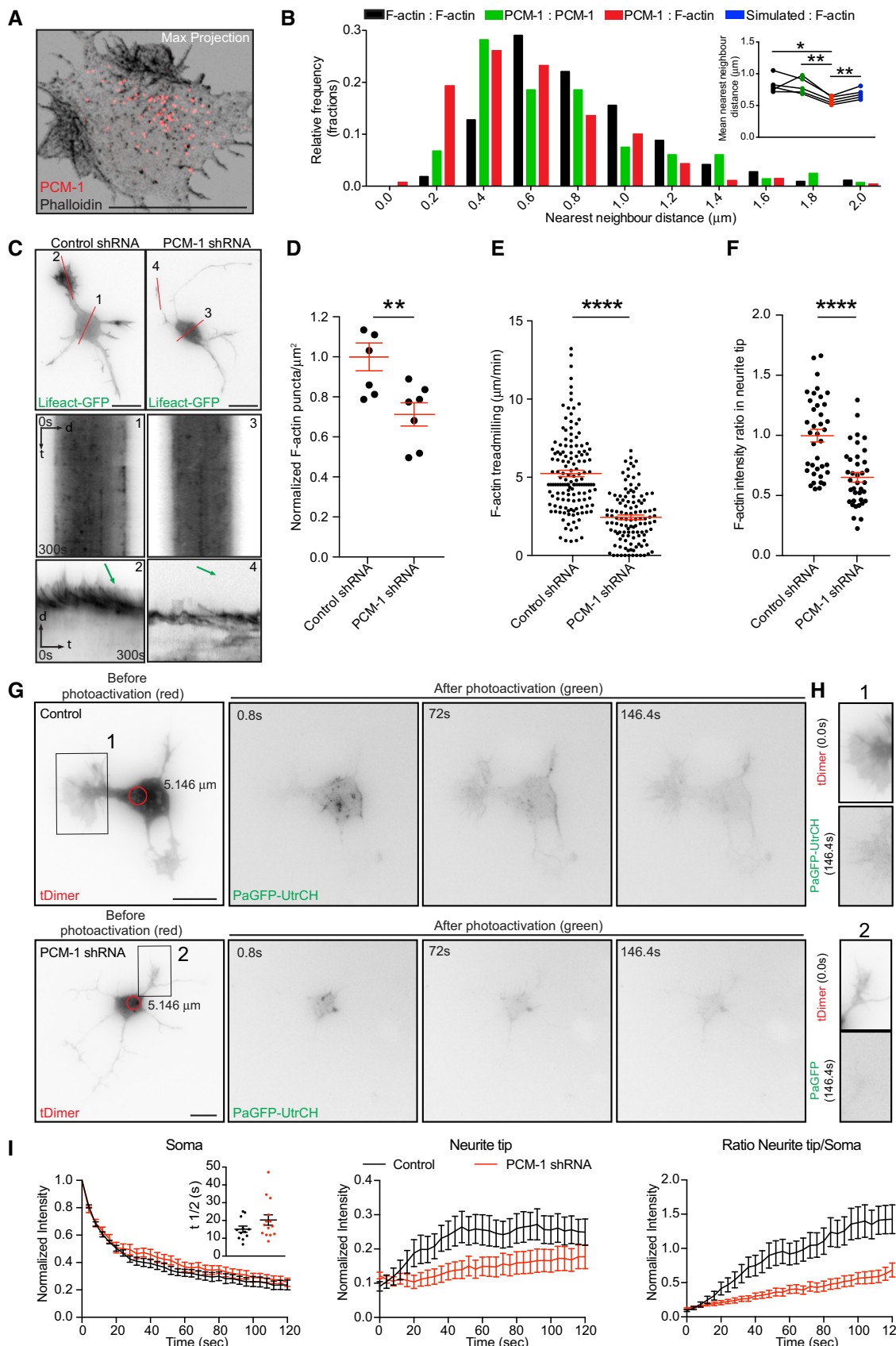

**Figure 6.  PCM-1 intermingles with F-actin puncta and is essential for the maintenance of F-actin in the soma and growth cones.**

A   Confocal Max projection of a stage 2 neuron stained with PCM-1 antibody and phalloidin showing polarized and intermingled PCM-1 and F-actin puncta.
B   Nearest neighbor analysis showing frequency distribution of distance between F-actin: F-actin puncta (black bars), PCM-1: PCM-1 puncta (green bars), and PCM-1: F-actin puncta (red bars). Inset: paired mean nearest neighbor distance values of F-actin:F-actin (0.827 ± 0.058), PCM-1:PCM-1 (0.811 ± 0.057), PCM-1:F-actin (0.584 ± 0.026) and simulated:F-actin (0.683 ± 0.036) puncta in the soma from $n$ = 5 cells, mean ± SEM. Paired $t$-test, *$P$ < 0.05, **$P$ < 0.01.
C   DIV I cortical neurons transfected with either control or PCM-1 shRNA together with Lifeact-GFP. Kymographs are obtained from lines marked as 1 and 3 for soma of control and PCM-1 shRNA, respectively. Lines 2 and 4 for growth cones of control and PCM-1 shRNA, respectively. Green arrows on the kymographs indicate the slope of F-actin treadmilling in the neurite tips.
D   Density of somatic F-actin puncta of stage 2 cells from control condition = 1.000 ± 0.0690, PCM-1 shRNA condition = 0.7122 ± 0.05809; **$P$ = 0.0078 by unpaired Student's $t$-test. Mean ± SEM; $n$ = 7 cells each for control and PCM-1 shRNA from at least three different cultures.
E   F-actin treadmilling speed (μm/min) in growth cones (or neurite tips from PCM-1 shRNA-expressing cells) of stage 2 cells from control condition = 5.2452 ± 0.2064; PCM-1 shRNA condition = 2.4402 ± 0.1543; ****$P$ < 0.0001 by unpaired Student's $t$-test. Mean ± SEM; $n$ = 10 cells for control and PCM-1 shRNA conditions, from at least three different cultures.
F   F-actin intensity ratio in growth cones of control condition = 0.9979 ± 0.0526, neurite tips of PCM-1 shRNA condition = 0.6513 ± 0.0401. ****$P$ < 0.0001 by unpaired Student's $t$-test. Mean ± SEM; $n$ = 10 cells for control, $n$ = 8 cells for PCM-1 shRNA groups from at least two different cultures.
G   DIV1 mouse cortical neurons *in utero* electroporated at E15 (cultured at E17) with PaGFP-UtrCH, tDimer, and control (upper panel) or PCM1-shRNA (lower panel) photoactivated in the soma with 405 nm laser (red circle with a diameter of 5.146 μm).
H   Neurites from cells in (G; insets 1, 2) show the reach of the photoactivated signal at the end of the time lapse (146 s).
I   Left panel: normalized intensity values in the photoactivated area of PaGFP-UtrCH expressing control and PCM-1 KD cells. Inset graph: half-time ($t\frac{1}{2}$) values for control condition = 15.12 ± 1.718 ($n$ = 12), PCM-1 KD condition = 20.28 ± 2.898 ($n$ = 14). $P$ = 0.1555 by unpaired Student's $t$-test. Middle panel: photoactivated signal in growth cones over time relative to the average initial signal from illuminated area for PaGFP-UtrCH expressing control and PCM-1 KD cells. Right panel: Growth cone to soma photoactivated signal intensity ratio of PaGFP-UtrCH expressing control and PCM-1 KD cells. All panels: mean ± SEM; $n$ = 12 cells for control condition, $n$ = 14 cells for PCM-1 KD condition, from at least two different cultures.

Data information: Scale bar: 10 μm.

---

between F-actin puncta-PCM-1 = 0.584 ± 0.019 μm; Fig 6A and B). Accordingly, neurons transfected with PCM-1-GFP showed PCM-1-GFP granules surrounding and "touching" somatic F-actin puncta (Appendix Fig S10A; Movie EV12).

To further test whether PCM-1 and somatic F-actin organization are interrelated, we treated neurons (24 h after plating) with cytochalasin D (1 μM for 3 h) or Jasplakinolide (500 nM for 4 h). We found that polarized F-actin structures induced by cytochalasin D or Jasplakinolide treatment are accompanied by PCM-1 particles (Appendix Fig S10B and C). Interestingly, when cells were co-treated with cytochalasin D (1 μM) and nocodazole (7 μM) for 3 h, disperse F-actin clusters (96.97%, 66 cells from at least three different cultures) associated with PCM-1 particles are formed (Appendix Fig S10D). These data indicate that somatic F-actin organization is linked to PCM-1 and microtubules integrity.

To probe the involvement of PCM-1 more specifically, we took advantage of *in utero* electroporation to introduce a PCM-1 shRNA construct to silence PCM-1 expression in cortical neurons and neuronal progenitors [23,24]. We examined the role of PCM-1 in F-actin dynamics and neurite outgrowth of cultured developing neurons and neurons differentiating in the developing cortex. PCM-1 down-regulation in cultured neurons led to the formation of long and thin neurites (Fig 6C; Appendix Fig S11A–D), similar to the well-known effect induced by pharmacological F-actin disruption using cytochalasin D [40]. Additionally, we tested whether PCM-1 down-regulation or F-actin disruption similarly affect neuronal differentiation in the developing cortex. We electroporated *in utero* control shRNA or PCM-1 shRNA, together with Venus, and DeAct plasmid—which impairs F-actin dynamics [41]—at E15 and analyzed the neuronal morphology at E18 *in situ*. Importantly, we found that down-regulating PCM-1 in the developing cortex and disrupting F-actin in newly born neurons promotes neurite elongation in a similar manner (Appendix Fig S11E–H). This suggests that PCM-1 down-regulation impairs F-actin dynamics and thus boosts neurite outgrowth.

In extension of these findings, we observed a direct effect of PCM-1 down-regulation on F-actin dynamics with a reduced total number of F-actin puncta in the cell body detected with Lifeact-GFP (Fig 6C and D) or Phalloidin (Appendix Fig S12A and B). Furthermore, PCM-1 down-regulation significantly decreased the F-actin treadmilling speed (Fig 6C and E) as well as the relative F-actin levels in neurite tips (Fig 6C and F). On this regard, we performed photoactivation of PaGFP-UtrCH in cortical neurons expressing PCM-1 shRNA and found that PCM-1 down-regulation drastically reduced the soma to neurite tips translocation of photoactivated signal (Fig 6G–I; Movie EV13). Of note, the effects of PCM-1 knock-down on F-actin puncta density in the soma, F-actin treadmilling speed and F-actin intensity in the neurite tips were reversed when an RNAi-resistant plasmid, chicken-PCM-1-GFP, was transfected along with Lifeact-RFP and PCM-1-shRNA (Appendix Fig S12C–F).

Thus, suggesting that PCM-1 down-regulation affects the amount of somatic F-actin, which is produced to modulate neurite outgrowth. Altogether our results show that PCM-1 regulates somatic F-actin dynamics and that somatic actin polymerization has an effect on growth cone dynamics.

**Formins determine the number of somatic F-actin puncta**

It was previously shown that axons of mature neurons contain F-actin "hotspots" which are the source of dynamic actin trails [42,43]. Importantly, Ganguly *et al* [42] showed that the activity of axonal F-actin "hotspots" correlates with stationary endosomes and are Formins-dependent. Disruption of endomembrane with BFA affected F-actin "hotspots" [42]. We tested whether somatic F-actin puncta are similar to the axonal F-actin "hotspots" reported in mature neurons. Using specific actin nucleator inhibitors (SMIFH2 and CK666), we were able to show that the somatic F-actin puncta are Formin- and not Arp2/3-dependent (Fig 7A–C) as reported for mature axons. Treatment with BFA, however, did not affect the number of somatic F-actin puncta (Appendix Fig S13A–C). Experiments with BFA (Appendix Fig S13A–C) and CK666 (Arp 2/3 inhibition; Fig 7A–C) treatments exclude the involvement of endosomal integrity and nucleation-promoting factors (NPFs, such as WASp,

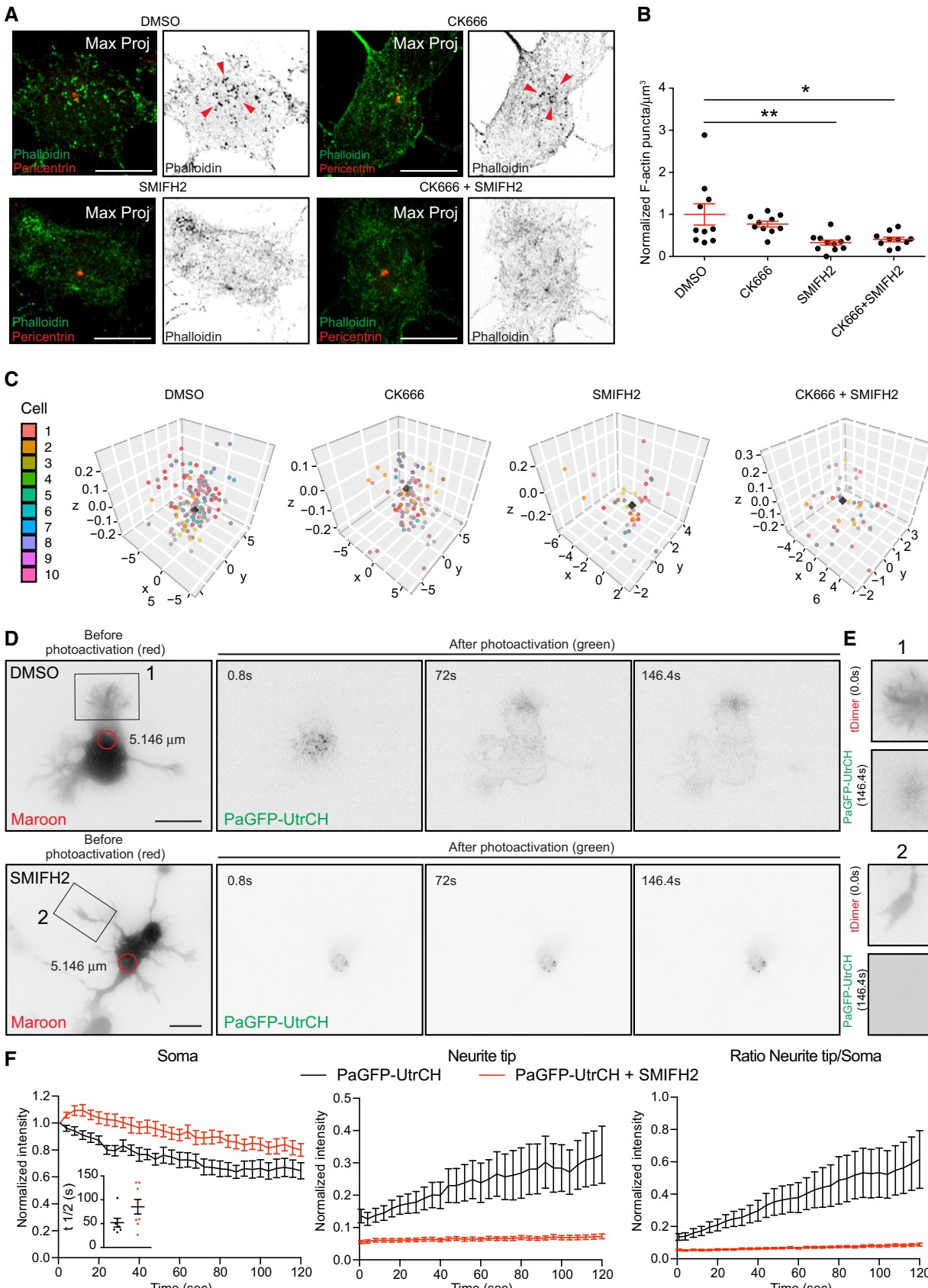

**Figure 7.**

◄

**Figure 7. Somatic F-actin puncta are Formin but not Arp2/3 dependent.**

A  DIV 1 neurons were incubated for two hours with DMSO, 50 μM CK666, 25 μM SMIFH2, or 50 μM CK666 + 25 μM SMIFH2. Cells were fixed and stained with anti-pericentrin antibody and Phalloidin.

B  Total number of cytosolic F-actin puncta per μm³ from cells treated with DMSO = 1.000 ± 0.2559, CK666 = 0.7740 ± 0.0693, SMIFH2 = 0.3308 ± 0.0605, and CK666 + SMIFH2 = 0.4067 ± 0.0554. *P* = 0.0036 by one-way ANOVA, *post hoc* Dunnett's test; *P < 0.05, **P < 0.01. Mean ± SEM; *n* = 11 for SMIFH2, *n* = 10 cells each for DMSO, CK666, and SMIFH2 + CK666 groups from at least two different cultures.

C  3D graphs show the distribution of cytosolic F-actin puncta around the centrosome in DMSO, CK666, SMIFH2, and CK666 + SMIFH2-treated cells. The coordinates of the centrosome and F-actin puncta are obtained from the confocal images. The coordinates of centrosomes from all the cells, indicated as a black cube, are positioned at the center (*x*, *y*, *z* = 0) and the color-coded F-actin puncta are plotted with respect to the position of the centrosome from the respective cell.

D  PaGFP-UtrCH and mMaroon1 co-transfected DIV1 rat hippocampal neuron treated with DMSO (upper panel) SMIFH2 (37.5 μM for 2 h; lower panel) photoactivated in the soma with 405 nm laser (red circle with a diameter of 5.146 μm).

E  Neurites from cells in (D; insets 1, 2) show the reach of the photoactivated signal at the end of the time lapse (146 s).

F  Left panel: normalized intensity values in the photoactivated area (soma) of DMSO- and SMIFH2-treated PaGFP-UtrCH expressing cells. Inset graph: half-time (*t*½) values in seconds for DMSO-treated cells = 52.00 ± 9.317 (*n* = 7), SMIFH2-treated cells = 85.33 ± 15.17 (*n* = 8). *P* = 0.0942 by unpaired Student's *t*-test. Middle panel: photoactivated signal in the neurite tip over time relative to the average initial signal from illuminated area for DMSO- and SMIFH2-treated PaGFP-UtrCH expressing cells. Right panel: neurite tip to soma intensity ratio of photoactivated for DMSO and SMIFH2-treated PaGFP-UtrCH expressing cells. All panels: mean ± SEM; *n* = 9 for DMSO and *n* = 13 cells for SMIFH2-treated PaGFP-UtrCH expressing cells, from at least two different cultures.

Data information: Scale bar: 5 μm (A), 10 μm (D).

Scar/WAVE, WASH, WHAMM/JMY) [44,45] in the organization of somatic F-actin, respectively. However, these findings do not entirely rule out the role of endosomal activity in the somatic F-actin organization.

Finally, we tested whether the Formins inhibitor SMIFH2 has an influence on somatic/neurite tip translocation of photoactivated UtrCH signal. Formins inhibition decreased the amount of photoactivated signal found in neurite tips upon 405 nm laser illumination in the soma (Fig 7D–F and Movie EV14). These results suggest some similarities between somatic F-actin puncta and F-actin "hotspots" in mature axons. We could only detect F-actin comet-like structures emanating from the somatic F-actin puncta (Fig 4B). However, in mature axons, actin trails emerging from F-actin "hotspots" were described. Thus, more studies need to be done to address the differences between the F-actin organization in the soma and axons.

## Discussion

Collectively, our results indicate that (i) F-actin in the cell body organizes around the centrosome and (ii) somatic F-actin is released toward the cell periphery, thus affecting growth cone behavior. To our knowledge, the neuronal F-actin organization described here is a novel cellular mechanism to sustain neuronal development. Our data suggest that F-actin content in growth cones has an inhibiting effect on neurite growth and this novel somatic F-actin organization sustains the growth cone organization. Accordingly, the distribution of the activated or converted somatic signal in our photoactivation/photoconversion paradigms is enriched preferentially in shorter neurites (Figs 2E and EV2F). More work, however, should be done to delineate the negative effect of neurite tip F-actin on neurite growth. Although our data do not clarify the molecular mechanism by which somatic F-actin is delivered toward the cell periphery, our results suggest that microtubule organization is relevant for somatic F-actin delivery to growth cones (Figs 5F and G, and EV5E and F).

Mechanistically, we show that this somatic F-actin organization in neurons relies on the presence of PCM-1. PCM-1 intermingles with somatic F-actin aster-like structures, which concentrate near the centrosome. Our time-lapse analysis further corroborates this PCM-1/somatic F-actin organization/vicinity. Moreover, our data

suggest that microtubules might be the link between centrosome and PCM-1/F-actin somatic puncta organization. Disruption of microtubules affected the distribution of PCM-1 and F-actin. Finally, and importantly, our data show that PCM-1 down-regulation affects F-actin dynamics in neurite tips of developing neurons and hence neuronal differentiation *in vitro* and *in vivo*. To our knowledge, this is the first time PCM-1 has been associated with F-actin dynamics in neurons and we are the first to show that PCM-1-dependent somatic F-actin organization has a direct effect on distal growth cones behavior.

However, our data do not show how PCM-1 regulates the somatic F-actin polymerization. Farina *et al* [6] described that centrosomal F-actin polymerization is Arp2/3 dependent. In contrast, we found that formins but not Arp2/3 contribute to the somatic F-actin organization in neurons, as shown for axonal F-actin organization of mature neurons [42,43]. Furthermore, a novel actin network was described, which mediates long-range vesicle transport and asymmetric oocyte division [46,47]. This actin network relies on actin nucleation factors such as Spire1, Spire2, and Formin-2. Thus, the mechanism described by Farina *et al* [6] may not be the same in all cell types. Further experiments need to be performed in neurons to ensure a deep understanding of the nature of somatic F-actin in neurons described here. In summary, we believe our data will pave the way to future important contributions oriented to understand F-actin organization and dynamics in developing neurons.

## Material and Methods

### RNAi and fluorescent protein constructs

We previously reported the Venus, mCherry, control shRNA, PCM-1 shRNA, and chicken-PCM-1-GFP plasmids [23,24]. A pSilencer vector containing a random sequence hairpin insert was used as a control for the shRNAs [23,24]. M. Harterink (from the laboratory of C. Hoogenraad, Utrecht University) kindly provided the DeAct-SpvB plasmid (that was cloned into the pGW2-GFP vector) [41]. The Centrin2-KR fusion protein was generated by cloning the Centrin2 cDNA in-frame with KillerRed, using the BamHI and XhoI cloning

sites of the pKillerRed-N vector (Evrogen) [24]. P. Gordon-Weeks kindly provided Drebrin-YFP (Addgene plasmid # 40359); [37] and Drebrin-S142D-YFP (Addgene plasmid S142D Drebrin-YFP # 58336); [35]. M. Davidson provided Lifeact-mEos3.2 (Addgene plasmid mEos3.2-Lifeact-7 # 54696), mEos3.2 (Addgene plasmid mEos3.2-C1 #54550), actin-mEos4b (Addgene plasmid mEos4b-actin-C-18 #57500), and Lifeact-RFP (Addgene plasmid mTagRFP-T-Lifeact-7 #54586) plasmids. F. Bradke (DZNE, Bonn) kindly provided the Lifeact-GFP plasmid. M. Kneussel (ZMNH, UKE) kindly provided the EB3-GFP and X. Chai (from the laboratory of M. Frotscher, ZMNH, UKE) kindly gave us the Cofilin-GFP and Cofilin-S3E-GFP constructs [48]. Cofilin-S3E-RFP was a gift from James Bamburg (Addgene plasmid # 50858; http://n2t.net/addgene:50858; RRID:Addgene_50858). Thomas Oertner (ZMNH, UKE) kindly provided the tDimer (pAAV-CAG-tDimer) plasmid. mMaroon1 (pcDNA3.1-mMaroon1) plasmid was provided by Michael Lin (Addgene plasmid # 83840; http://n2t.net/addgene: 83840; RRID:Addgene_83840). PaGFP was a gift from Karel Svoboda (Addgene plasmid # 18697; http://n2t.net/addgene:18697; RRID: Addgene_18697). PaGFP-UtrCH was a gift from William Bement (Addgene plasmid # 26738; http://n2t.net/addgene:26738; RRID: Addgene_26738).

## Animal experiments

Animal (rat and mouse) experiments were performed according to the German and European Animal Welfare Act and with the approval of local authorities of the city-state Hamburg (Behörde für Gesundheit und Verbraucherschutz, Fachbereich Veterinärwesen) and the animal care committee of the University Medical Center Hamburg-Eppendorf.

## Preparation of hippocampi and cortices from rat and mouse embryos

Pregnant rats and mice were anesthetized with $CO_2/O_2$ and then euthanized before taking the embryos out from their uteri. Embryos were then decapitated, and heads were collected in petri dishes kept on ice. After opening the skulls, brains were collected in petri dishes with HBSS on ice. Hemispheres were separated, meninges were carefully stripped away, and hippocampi/cortices were dissected.

## Hippocampal neuronal cultures and transfections

Isolated hippocampi (from E18 embryos) were triturated in 1xHBSS (Invitrogen) after digestion by papain and DNase for 10 min at 37°C (Worthington). Transfections were performed using the Amaxa nucleofector system following the manufacturer's manual. $5 \times 10^6$ cells and 3 µg of DNA mix was used for each transfection. Lifeact-GFP, Lifeact-RFP, Centrin2-RFP, tDimer, mMaroon1, PaGFP, PaGFP-UtrCH, Lifeact-mEos3.2, mEos3.2, actin-mEos4b, Drebrin-YFP, Drebrin S142D-YFP, Cofilin S3E-GFP, Cofilin-S3E-RFP, Centrin2-KR, EB3-mCherry, and EB3-GFP, plasmids were used for hippocampal neuronal transfections. The final concentration for each plasmid was 1 µg, except for Lifeact-GFP, Lifeact-RFP, and tDimer. We used 0.5 µg of Lifeact-GFP, Lifeact-RFP, and 0.3 µg of tDimer plasmids. In some cases, empty pcDNA 3.1 was used to make up to 3 µg of DNA per each transfection mix as per the manufacturer recommendation. After electroporation, neurons were plated on poly-L-lysine

coated glass coverslips (for immunostaining), on glass-bottomed dishes (ibidi, for live imaging) or tissue culture chambers (Sarstedt, for live imaging) in Neurobasal/B27 medium (Invitrogen) and were maintained in culture for 24 h at 37°C with 5% $CO_2$ before use.

## In utero electroporation

Pregnant C57BL/6 mice with E15 embryos were first administered with pre-operative analgesic, buprenorphine (0.1 mg/kg), by subcutaneous injection. After 30 min, mice were anesthetized with isoflurane (4% for induction, 2–3% for maintenance) in oxygen (0.5–0.8 l/min for induction and maintenance). Later, uterine horns were exposed and plasmids mixed with Fast Green (Sigma) were microinjected into the lateral ventricles of embryos. Five current pulses (50 ms pulse/950 ms interval; 35–36 V) were delivered across the heads of embryos. After surgery, mice were kept in a warm environment and were provided with moist food containing post-operative analgesic, meloxicam (0.2–1 mg/kg), until they were euthanized for collection of the brains from the embryos. The brains were either used for cortical cultures or cortical slices.

For cortical cultures, we introduced PCM-1 shRNA (with or without PCM-1-GFP) or control shRNA plasmids together with Lifeact-GFP, Lifeact-RFP, PaGFP-UtrCH in combination with tDimer or Venus plasmids into brain cortices at embryonic day 15 (E15) and isolated cortical neurons at E17. The concentration of shRNA (control or PCM-1-shRNA), PCM-1-GFP plasmids injected was 3-fold higher than that of the Lifeact-GFP, Lifeact-RFP, PaGFP-UtrCH, or Venus plasmids. We used 1.5 µg/µl for shRNA (control or PCM-1-shRNA), PCM-1-GFP and 0.5 µg/µl for Lifeact-GFP, Lifeact-RFP, PaGFP-UtrCH, and Venus plasmids, 0.3 µg/µl of tDimer. Neurons were cultured *in vitro* for an additional 24 h and were prepared for time-lapse imaging or pharmacological treatments or photoactivation experiments.

For cortical slices, we injected Centrin2-RFP together with Lifeact-GFP or PCM-1 shRNA or control shRNA plasmids together with Venus or DeAct-SPvB together with mCherry into brain cortices at embryonic day 15 (E15) and brains were collected at E18. The concentration of Centrin2-RFP, shRNA (control or PCM-1-shRNA), DeAct-SpvB plasmid injected was twofold to threefold higher than that of the Lifeact-GFP, Venus, or mCherry plasmids. We used 1.5 µg/µl of shRNA (control or PCM-1-shRNA), 1.0 µg/µl for Centrin2-RFP, DeAct-SpvB, and 0.5 µg/µl for Lifeact-GFP, Venus, and mCherry.

## Cortical cultures

Neurons were transfected by *in utero* electroporation at E15 and transfected cortices were dissected two days later (as explained above). Isolated cortices were triturated in 1xHBSS (Invitrogen) containing papain and DNase at 37°C (Worthington, [24]. Neurons were plated on poly-L-lysine coated glass-bottomed dishes (ibidi) in Neurobasal/B27 medium (Invitrogen) and were maintained in culture for 24 h before time-lapse imaging.

## Cortical slices

Embryonic brain cortical regions were transfected via *in utero* electroporation at E15. Lifeact-GFP and Centrin2-RFP were used to inspect F-actin distribution, PCM-1 shRNA, or DeAct were used to perturb the F-actin organization. The brains collected from E18

embryos were then post-fixed in 4% paraformaldehyde (PFA) overnight at 4°C and later moved to 30% sucrose (in PBS) until they were completely sunk. Brains were then embedded in Tissue Tek OCT compound and stored at −80°C until they were sectioned to 50-μm slices using a cryostat.

## STED microscopy

All STED images were acquired on a Leica TCS SP8 gated STED microscope equipped with a pulsed 775 nm depletion laser (80 MHz) and a pulsed white light laser (WLL) for excitation. The microscope was covered by an incubation chamber Black i8 2000 LS (PeCon GmbH, Erbach, Germany) fitted to the Leica microscope stand DMI 6000AFC. The chamber can be temperature controlled with the PeCon Temp Controller 2000-2 and Heating Unit 2000. For acquiring images, Leica Objective HC APO CS2 100×/1.40 Oil was used.

## STED imaging in fixed cells

Neurons stained for actin with Atto 647N-Phalloidin (1:40; Stock 10 nM, Sigma-Aldrich, #65906) and pericentrin with anti-rabbit Atto 594 (1:200; Sigma-Aldrich) or anti-rabbit Abberior Star 580 (1:200; Abberior GmbH Gottingen, Germany) were embedded in Mowiol and excited via the WLL at 650 and 561 nm, respectively. Emission was acquired between 660 and 730 nm for Atto 647N and 580–620 nm for Atto 594. The detector time gates for both channels were set from 0.5–1 ns to 6 ns. Both dyes were depleted with 775 nm. Respective confocal channels use the same settings as STED channels, except for a reduction of excitation power. Detection time gates were set from 300 ps to 6 ns for both channels. The Format for all images was set to 1,024 × 1,024. Optical zoom of five resulted in a voxel size of 23 nm for x-y and 100–160 nm for z. Images where then taken with 600 lines per second and line averaging of 8.

## STED imaging in live cells

For live STED microscopy of DIV 1 hippocampal primary neurons, 250 nM or 500 nM of SiR-actin (tebu-bio GmbH, Offenbach, Germany via Spirochrome AG, Switzerland, [26]) was directly added to the conditioned medium, and cells were imaged after 1.5–3 h (for 250 nM) and 5 h (for 500 nM).

Time lapses (1.7-s interval) were imaged in pre-carbonated extracellular solution at 37°C. SiR-actin was excited using a 633 nm light from the WLL, and fluorescence signals were detected via internal Leica Hybrid Detectors in the range of 650–750 nm. The detection time gate was set to 0.3–6.0 ns. Depletion was done at 775 nm at low depletion power of 30%. Application of a zoom of 5 and a format of 1,024 × 256 resulted in a pixel size in x-y of 23 nm. Scan speed was set to 600 Hz by applying four times line averaging.

To improve image quality, raw data of STED images were deconvolved using the STED package Huygens Professional (SVI, v 15.10) as follows: To calculate the theoretical point spread function (PSF), the optical microscopic parameters provided with the Leica lif-file were used. Within the *Deconvolution wizard*, time-lapse images were subjected to the automatic background calculation of Huygens software. For deconvolution, the *signal-to-noise ratio* was set to 15. The *Optimized iteration mode of the CMLE* was applied, until it reached a *Quality threshold* of 0.05.

## SMLM, Photoconversion, and FRAP

Photoconversion were performed on a Visitron Systems VisiScope TIRF/FRAP imaging system based on a Nikon Ti-E equipped with a Nikon CFI Apo TIRF 100×, 1.49 NA oil objective, a back focal TIRF scanner for suppression of interference fringes (ILas-2, Roper Scientific France/PICT-IBiSA, Institut Curie), and controlled via VisiView software. 405, 561, 488, and 647 nm laser lanes were used for illumination and activation of respective fluorophores. Fluorescence was detected through an ET 405/488/561/640 Laser Quad Band filter with The ORCA-Flash 4.0 LT sCMOS camera. Cells were maintained at 37°C in a temperature, $CO_2$, and humidity-controlled environmental chamber for all live-imaging experiments.

## SMLM

For SMLM imaging of F-actin, primary neurons were fixed at DIV1 and immonostained against pericentrin. Phalloidin conjugated to Alexa Fluor 647 (Life Technologies) was added to PBS in a 1:50 dilution and incubated at room temperature (RT) for 3 h. After extensive washing, samples were post-fixed for 10 min with 4% PFS + 4% sucrose in PBS, washed again with NT buffer (50 mM Tris–HCl pH 8, 10 mM NaCl). Samples were sealed using cavity coverslips filled with Buffer TN containing 10% glucose, 10 mM freshly prepared mercaptoethylamine (pH adjusted to 8 with KOH; Sigma), 40 μg/ml catalase (Sigma), and 0.5 mg/ml glucose oxidase (Sigma).

For SMLM imaging of Alexa Fluor 647, the sample was continuously illuminated with 640 nm light. In addition, the sample was illuminated with 405 nm light at increasing intensity to keep the number of fluorophores in the fluorescent state constant. Between 6,000 and 10,000 frames were recorded per acquisition with an exposure time of 50 ms. Reconstruction of SMLM images was done as described previously using custom-written software (Detection of Molecules; [1,49]).

## Photoconversion experiments

Photoconversion of Lifeact-mEos3.2mEos3.2 or mEos4b constructs expressed in primary rat hippocampal neurons was performed using the ILas2 system. Photoconversion was achieved using the 405 nm laser with 1% laser power. For the photoconversion in the soma, the circular area of illumination was in the range of 520 pixels (with a diameter of 2.1 μm) to 1,700 pixels (with a diameter of 7.2 μm). For neurite tip photoconversion, the illuminated circular area was in the range of 1,240 pixels (with a diameter of 5.2 μm) to 2,680 pixels (with a diameter of 11.3 μm). The time of 405-laser illumination was always set to 1 ms per pixel (1 pixel = 65 nm). For recording green and red fluorescence, cells were sequentially illuminated with 488 nm laser (2% power) and 561 nm laser (5% power) for 400 ms. Three frames each of green and red fluorescence were recorded before and 150 frames (800 ms/frame, i.e., 120 s) each were recorded after the photoconversion. The green and red fluorescence was detected through the 540 and 620 nm emission filters, respectively.

For photobleaching evaluation experiments performed on 4% PFA-fixed primary neurons expressing mEos3.2 or mEos4b-tagged constructs, all the imaging settings were the same as above but with

the following exceptions. Photoconversion was performed using the 405 nm laser with 20% laser power. The area of 405 nm laser illumination was always 1,218 pixels (with a diameter of 5.146 μm). For recording green and red fluorescence, cells were sequentially illuminated with 488 nm laser (20% power) and 561 nm laser (50% power) for 400 ms.

## Photoactivation experiments

Photoactivation of PaGFP and PaGFP-UtrCH was performed either in the soma or neurite tip of primary rat hippocampal or mouse cortical neurons using the ILas2 system. To identify transfected neurons prior to photoactivation, the cultures used for this experiment were always co-transfected with tDimer or mMaroon1 together with PaGFP-UtrCH. To illustrate the neuronal morphology, an image with tDimer or mMaroon1 signal was captured before the photoactivation with 561 nm or 640 nm lasers, respectively. Photoactivation of PaGFP-UtrCH was achieved using the 405 nm laser with 50% laser power. The photoactivated circular region was always between 1,218 pixels (with a diameter of 5.1 μm) or 1,240 pixels (with a diameter of 5.2 μm); the 405 nm laser illumination time was set to 2 ms per pixel (1 pixel = 65 nm). Cells were illuminated with 488 nm laser (5% power) for 800 ms to record one frame. Three frames were imaged before, and 150 frames (800 ms/frame, i.e., 120 s) were recorded after the photoactivation.

For photobleaching evaluation experiments performed on 4% PFA-fixed primary neurons expressing PaGFP-tagged constructs, all the imaging settings were the same as above. The area of 405 nm laser illumination was always 1,218 pixels (with a diameter of 5.146 μm).

## FRAP experiments

Photobleaching of GFP signal was performed either in the soma or in neurite tip of Lifeact-GFP expressing primary rat hippocampal neurons using the ILas2 system. Photobleaching was achieved using the 405 nm laser with 100% laser power. The photobleached circular region was 720 pixels (with a diameter of 3.04 μm); the 405 nm laser illumination time was set to 1 ms per pixel (1 pixel = 65 nm). Cells were illuminated with 488 nm laser (8% power) for 75 ms to record one frame (70–75 nm TIRF penetration depth). Three frames were imaged before, and 188 frames (75 ms/frame, i.e., 14.1 s) were recorded after photobleaching.

## Pharmacological treatments

Pharmacological compounds were directly added to DIV1 rat hippocampal or mouse cortical neurons in culture. The treated cells were either used for PFA (4%) fixation for immunostaining or time-lapse live imaging (Table 1).

## F-actin extension frequency analysis (from STED live images)

Stage 2 and early Stage 3 primary hippocampal neurons incubated with 250 nM SiR-actin for 1.5–3 h were used for analysis. The frequency of F-actin extensions radiating from the F-actin puncta in approximately 1-min duration (36 frames with an interval of 1.7 s = 61.2 s) was manually measured using the ROI manager (ImageJ).

Table 1. Information on the pharmacological compounds used in the study.

| Compound | Concentration | Related figures and incubation times |
|---|---|---|
| Brefeldin A (BFA) | 10 mg/ml | Fig EV3C: Fixed after 12 h |
| | 1 mg/ml | Appendix Fig S13: Fixed after 12 h |
| CK666 | 50 μM | Fig 7A–C: Fixed after 2 h |
| Cytochalasin D (CytoD) | 1 μM | Fig EV3A and D–G: Live imaging between 45 min and 2 h |
| | | Fig EV3B and C: Fixed after 12 h |
| | | Appendix Fig S10C and D: Fixed after 3 h |
| | | Appendix Fig S11A–D: Fixed after 24 h |
| Jasplakinolide (Jasp) | 500 nM | Appendix Figs S8A and S10B: Fixed after 4 h |
| | | Appendix Fig S8B: Live imaging after 4 h |
| | 300 nM | Appendix Fig S8C–F: Live imaging between 1.5 and 3 h |
| Nocodazole (Noc) | 7 μM | Figs 5F and G, and EV5E and F: Live imaging between 1.5 and 3 h |
| | | Appendix Fig S10D: Fixed after 3 h |
| SMIFH2 | 25 μM | Fig 7A–C: Fixed after 2 h |
| | 37.5 μM | Fig 7D–F: Live imaging between 2 and 3 h |

## Photoconversion/Photoactivation analysis

The diameter of 405 nm laser illuminated area for all the cells used for the analysis was between 5.239 and 7.182 μm. 561 nm channel time lapses of photoconverted cells were preprocessed with ImageJ [50,51] by subtracting the first empty frame from the remaining frames. For analysis, the area of photoconversion at either soma or neurite tips was selected as a ROI. Further ROIs were selected in the corresponding compartments (soma, neurite tip). For the soma, we tried to select an area avoiding the nucleus, for growth cones/neurite tips we selected an area, which remains within the central region, even during eventual movement. If the growth cone was moving too much, the region was cropped and aligned via ImageJ's template matching plugin to avoid moving artifacts. For plots, 0 was considered as the first frame after photoactivation/photoconversion. Average gray values over time were measured via the Time series Analyzer plugin. Initial gray values in the photoconverted area were normalized to 1; gray values in the "receiving" compartments were normalized to fractions of initial gray value at the photoconverted area. The ratio of the average neurite tip intensity to the average soma intensity was plotted over time for different experimental groups. For the sake of clarity, every 4-s were considered. Furthermore, the intensity ratio of neurite tip to soma after two minutes was compared to the intensity ratio of neurite tip to soma before photoconversion (green channel). The association of these parameters was determined via Pearson correlation. For the decay of signal in the photoconverted area in the soma $t\frac{1}{2}$ was determined via Graph-Pads fitted one-phase exponential decay equation. If the calculation of $t\frac{1}{2}$ lead to ambiguous values,

they were not considered for comparison. The same analysis was performed for photoactivated PaGFP-Utrch expressing cells.

### FRAP analysis

Fluorescence recovery after photobleaching in the somatic and growth cone areas of the primary hippocampal neurons expressing Lifeact-GFP was measured using the FRAP profiler and Intensity vs. Time Plot plugins (ImageJ). A simple ratio bleach correction was applied before analysis.

### Actin & PCM-1 nearest neighbor analysis

Actin- and PCM-1 puncta were counted manually within the three-dimensional soma space and marked in their estimated center with ImageJ's selection tools. Center coordinates were then exported via the ROI manager for further processing with R. For each coordinate, a distance matrix was calculated and filtered to find the nearest neighbor. Shortest distances of puncta to the closest object—within and between the actin and PCM-1 datasets—were collected and plotted as a relative frequency distribution. Additionally, the average minimum distance was reported.

### Immunocytochemistry

Rat hippocampal neurons grown on coverslips were fixed with 4% paraformaldehyde (PFA) at 37°C for 10 min and then permeabilized with 0.5% Triton X-100 for 10 min. Non-specific binding was blocked by incubation with 5% donkey serum in PBS for 60 min at RT, followed by specific primary antibody incubation: rabbit anti-pericentrin (Convance/dcs diagnostics, PRB-432C), rabbit anti-PCM-1 (kindly gifted by Andreas Merdes), mouse anti-GM-130 (BD Biosciences, 610823), or mouse anti-α-tubulin (Abcam, ab7291) was added for incubation for 180 min at RT. The respective anti-mouse or anti-rabbit Alexa Fluor −488 or −568 labeled secondary antibody was added for 60 min at RT. Primary and secondary antibodies were diluted in PBS with 2% donkey serum. After primary and secondary antibody incubation, three washing steps with PBS were performed. For F-actin labeling, cells on coverslips—after PFA fixation and permeabilization—were incubated with Phalloidin 488 from Cytoskeleton Inc. (for confocal imaging) or Phalloidin 647 (for pharmacology experiments, Invitrogen), followed by three PBS washes. In some experiments, Hoechst dye (1:10,000, Invitrogen) was added to stain nuclei. Coverslips were mounted onto slides using prolong gold (Invitrogen) and were stored protected from light.

### Epifluorescence imaging

Epifluorescence imaging was performed on an inverted Nikon microscope (Eclipse, Ti) with a 60× objective (NA 1.4). During time-lapse imaging, cells plated on a glass-bottomed dish (Ibidi) or a culture chamber (Sarstedt or Ibidi) were kept in an acrylic chamber at 37°C in 5% $CO_2$. Light intensity of each channel was normally set at 8, with an exposure time of 300–800 ms. Images were captured with a CoolSNAP HQ2camera (Roper Scientific) using NIS-Elements AR software (version 4.20.01 from Nikon Corporation).

### Intensity measurements

For CALI experiments, neurite tip areas were delineated with ImageJ's selection tools for the first frame of the acquired time lapse. Mean intensities from these selections were compared for cells before and after CALI treatment. Intensity values after CALI treatment were normalized to intensity values before CALI treatment. For ctrl shRNA, PCM-1 shRNA, and PCM-1 rescue, additionally the mean intensity within the soma was measured and the neurite tip to soma ratio was calculated.

### Confocal imaging

Images were taken using an Olympus FLUOVIEW FV1000 and Zeiss LSM 700 confocal laser scanning microscopes with 40× objectives (NA 1.3). Z-series images were acquired with a step size of 300 nm or 400 nm.

### Confocal F-actin puncta analysis

For centroid analysis shown in Fig 1A and B, Appendix Fig S1A and B, and Fig 7A–C, Appendix Fig S13, confocal images of DIV1 or DIV2 rat hippocampal neurons labeled with phalloidin (F-actin) and pericentrin antibody were used. The confocal images (with a *z*-stack spacing of 400 nm) were first thresholded using Auto Local Threshold v1.5 (ImageJ) with Phansalkar as the method of choice with the radius set according to the size of the cells and their F-actin puncta (radius 2–10, [52]). Despeckle command (ImageJ) was used to reduce noise. The plugin 3D Objects Counter v2.0 (ImageJ) [53] was applied on each image for automatic counting of objects as well as for defining the respective centroid coordinates. The counted objects were then manually compared to the original image, and false positives were removed. ImageJ's plugin Volumest was used for determination of the cell volume following the author's instructions (Merzin, Markko. "Applying stereological method in Radiology. Volume measurement". Bachelor Thesis. University of Tartu. 2008). Data were represented as normalized F-actin puncta/μm³ (in Fig 7B and Appendix Fig S13B). For cytosolic puncta in relation to the centrosome analysis, the volume of the nucleus was removed and all puncta were considered, if there was no space occupied between the puncta and the centrosome. The density in dependence of distance was visualized in ridge plots for all cells (in Fig 1B and Appendix Fig S1B). To show the underlying distribution, distance was normalized via *z*-score (values are centered around the mean and expressed as their distance to the mean in terms of standard deviation) to account for different cell sizes. The resulting comparison is therefore in terms of standard deviations from the mean (in Fig 1B and Appendix Fig S1B). For an overview coordinates of F-actin, puncta are color-coded for individual cells and superimposed in a 3D plot with cells aligned at the centrosome (in Figs 1B and 7C, Appendix Figs S1B and S13C).

In case of Appendix Fig S12A and B, confocal images of DIV 2 cortical neurons co-transfected with Venus along with control or PCM-1 shRNA plasmids via *in utero* electroporation were used. Confocal images with a *z*-stack spacing of 300 nm labeled with phalloidin (F-actin) and anti-PCM-1 antibody were used for analysis. Somatic F-actin puncta in control and PCM-1 KD cells were counted manually using ImageJ (NIH). Volumes of the somas were

measured using ImageJ (NIH) Volumest plugin as discussed above. Normalized F-actin puncta/μm$^3$ cell volume were plotted for both the groups.

### Analysis of somatic F-actin puncta and F-actin treadmilling in neurite tips

For analysis of F-actin puncta in the soma and F-actin treadmilling flow in neurite tips, 5-min time-lapse videos of Lifeact-GFP or Lifeact-RFP expressing rat hippocampal or mouse cortical neurons were used.

### Analysis of somatic F-actin puncta in living cells

For centroid analysis (shown in Fig EV1A and B), 5-min time lapse with 2-s intervals (151 stacks) of the cells that were transfected with Lifeact-GFP and EB3-mCherry was used. Soma area was selected and radially divided into 12 equal segments. In each segment, recognizable puncta were counted through 151 stacks. To characterize puncta blinking preference in the cell body, puncta sum of each segment for all stacks was made and after measuring the area of each segment within cell body, puncta density was calculated and plotted as rose diagrams. Regarding the "Quadrant" definition, the three segments covering the central microtubule organization area (judged based on the co-transfected EB3-mCherry signal) was considered as the first quadrant, namely Q1, followed by Q2, Q3, and Q4 clockwise (each quadrant covers 3 segments accordingly). To examine the blinking frequency of puncta in each quadrant, puncta number of each stack was plotted after converted into percentage by being divided by the total puncta number of all stacks within the same quadrant.

For analysis of stability and total number of F-actin puncta in living cells, the soma area of Lifeact transfected neurons from the 5-min time-lapse videos was marked and resliced with an output spacing of 1.0 pixel using ImageJ to create kymographs. Stability duration and total number of F-actin puncta present in the soma area were then analyzed via ImageJ's selection tools, adding selections to the ROI manager. Length of Lifeact labeled structures in the kymographs along the *y*-axis was used for defining stability duration with each pixel representing 2 s (151 frames for 5 min). Based on duration, Lifeact labeled structures were distinguished as unstable puncta: distinct Lifeact spots or vertical lines disappearing in 15 s or less; puncta with intermediate stability: vertical Lifeact lines, appearing for 16–240 s and long-lasting puncta: vertical Lifeact lines that are in the range of 241–300 s. The total number of puncta per μm$^2$ was obtained through the sum of unstable, intermediate and long-lasting F-actin puncta normalized to the area of soma marked for analysis.

### Analysis of F-actin treadmilling in neurite tips

Kymographs were generated from the neurite tips (from the 5-min time-lapse videos) of Lifeact (tagged with GFP or RFP) transfected rat hippocampal or mouse cortical neurons using the ImageJ Kymograph plugin (code written by J. Rietdorf & A. Seitz, EMBL Heidelberg). From the kymographs (generated by setting line width to 1), the slope of retrograde trajectories of F-actin was measured and average slope was represented in μm/min.

### Laser-induced chromophore-assisted light inactivation of Centrin2-KR

Rat hippocampal neurons obtained from E18 embryos were co-transfected with Centrin2-KR and either Lifeact-GFP or EB3-GFP and were maintained in culture for 24 h before experiments. Neurons at stage 2 or early stage 3 were selected. After recording a 5-min time lapse (2 s interval) under 100× objective (CFI Apo TIRF, oil, NA 1.49), the neuron selected was illuminated locally at the centrosomal site with FRAP 561 nm laser at 7% laser power for 1.5 s in the region of interest (~ 2 μm$^2$) to inactivate Centrin2-KR and thus to disrupt the centrosome. After 2–3 h of recovery (at 37°C with 5% $CO_2$), the same cells were recorded for another 5 min. Control experiments were performed with the same settings, but cells were illuminated at a somatic site away from the centrosome.

### EB3 comets quantifications

For CALI cells, the soma area of EB3-GFP expressing cells was selected. Comets were tracked manually throughout the first 2 min using ImageJ's ROI manager and selection tools. Number of EB3 comets was normalized by area and was reported as average number of comets per μm$^2$ * minutes.

### Neurite length measurement and neurite terminals analysis

Rat hippocampal neurons transfected with tDimer alone or together with Drebin-YFP, Drebrin-S142D-YFP, Cofilin-GFP, Coilin-S3E-GFP were used for neurite length analysis (in Appendix Fig S9D and E). Mouse cortical neurons co-transfected with Venus and control or PCM-1-shRNA via *in utero* electroporation, cells from the control group treated with cytochalasin D were used for neurite length and terminals analysis (in Appendix Fig S11A–D). For neurons *in situ*, mouse brains transfected either with PCM-1-shRNA and Venus or DeAct-SpvB and mCherry were used for neurite length and terminals analysis (in Appendix Fig S11E–H).

### Neurite length analysis

In each cell total length of all neurites, length of the longest neurite and length of other neurites were measured manually from all the mentioned conditions using ImageJ (NIH). In case of *in situ* analysis, only multipolar cells with three or more primary neurites were considered and traced via ImageJ's simple neurite tracer [54] to account for the 3D distribution.

Number of primary neurites with length greater than 60 μm per cell was counted manually from all the three conditions.

### Neurite terminals analysis

Total number of neurite terminals per cell was counted manually from all the three conditions.

### Image processing

Linear adjustment of brightness and contrast was performed on images using Photoshop CS or ImageJ.

Durga Praveen Meka *et al*

EMBO reports

## Statistical analysis

Statistical analysis was performed using the GraphPad Prism 6 software. Data shown in the graphs were collected from two to three independent experiments. The Student's *t*-test (two-tailed) was used to compare means of two groups, whereas analysis of variance (ANOVA) was used when comparing more than two groups. Asterisks *, **, *** and **** represent $P < 0.05$, 0.01, 0.001, and 0.0001, respectively. Error bars in the graphs always represent standard error of mean. Pearson correlation analysis was performed for showing a linear relationship between two sets of data. Z-score normalization was done for data sets with multiple populations (e.g., distance of puncta to centrosome for each cell) by centering values around the mean and dividing by the standard deviation. Resulting values are therefore expressed as standard deviations (σ) from the mean.

**Expanded View** for this article is available online.

## Acknowledgements

We thank Bas van Bommel (M. Mikhaylova's lab, ZMNH, Hamburg), A. Merdes (CNRS, Toulouse), M. Kneussel (ZMNH, Hamburg), M. Harterink (C. Hoogenraad's laboratory, Utrecht), T. Oertner, X. Chai (M. Frotscher's lab, ZMNH, Hamburg), and P. Soba (ZMNH, Hamburg) for antibodies, plasmid constructs, and equipment use. Thanks to I. Hermans-Borgmeyer and members of UKE-Hamburg animal facility, Hamburg for their help with animal experiments and O. Durak (Harvard University) for critical reading of the manuscript. M. Kreutz is supported by the Leibniz Foundation, Deutsche Forschungsgemeinschaft (Kr1879 5-1, 6-1; SFB 779 TPB8), BMBF Energi and JPND STAD. M. Mikhaylova is supported by grants from the Deutsche Forschungsgemeinschaft (DFG Emmy-Noether Programm (MI 1923/1-1) and FOR2419 (MI 1923/2-1)). O. Kobler is funded by the DFG Grant SCHE 132/18-1. F. Calderon de Anda is supported by Deutsche Forschungsgemeinschaft (DFG) Grants: FOR 2419, CA1495/1-1, and CA 1495/4-1; ERA-NET Neuron Grants (Bundesministerium für Bildung und Forschung, BMBF, 01EW1410, and 01EW1910), JPND Grant (Bundesministerium für Bildung und Forschung, BMBF, 01ED1806), and University Medical Center Hamburg-Eppendorf (UKE). DP. Meka is a co-applicant in the Deutsche Forschungsgemeinschaft (DFG) Grant CA 1495/4-1 for F. Calderon de Anda.

## Author contributions

FCA conceived the idea, supervised the project, and wrote the manuscript. FCA and DPM designed the study. BS, BZ, and DPM conducted all the cell culture. OK, MRK, MM, and DPM performed STED imaging. MM carried out the SMLM imaging. DE helped with the initial SMLM imaging. DPM performed all the photoconversion, photoactivation, and FRAP experiments; RS and FCA analyzed the data. DPM and MR performed all *in utero* electroporation surgeries; DPM, RS, and IS analyzed the data. BZ performed CALI experiments, BZ and RS and DPM analyzed the data. SK helped setting up the microscope for photoconversion, photoactivation, FRAP, and CALI experiments. RS and DPM performed all the confocal imaging. TK, IS, RS, and DPM performed F-actin puncta analysis. BZ and DPM performed the epifluorescence time-lapse imaging. DPM, RS, BZ, and SW analyzed the data. FCA, DPM, and CGD performed pharmacological experiments. FCA supervised DPM, BZ, RS, TK, IS, BS, MR, and MM supervised SK. All authors helped writing the manuscript.

## Conflict of interest

The authors declare that they have no conflict of interest.

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
