## [Review Process File · EMBO Reports]

Radial Somatic F-actin Organization Affects Growth Cone Dynamics During Early Neuronal Development

Durga Praveen Meka, Robin Scharrenberg, Bing Zhao, Oliver Kobler, Theresa König, Irina Schaefer, Birgit Schwanke, Sergei Klykov, Melanie Richter, Dennis Eggert, Sabine Windhorst, Carlos G. Dotti, Michael R. Kreutz, Marina Mikhaylova, Froylan Calderon de Anda

Review timeline:	Submission date:	18 January 2019
	Editorial Decision:	26 February 2019
	Revision received:	8 July 2019
	Editorial Decision:	28 August 2019
	Revision received:	6 September 2019
	Accepted:	27 September 2019

Editor: Deniz Senyilmaz-Tiebe

Transaction Report:

1st Editorial Decision

26 February 2019

Thank you for submitting your manuscript for consideration by EMBO Reports. It has now been seen by three referees whose comments are shown below.

As you can see, all referees express interest in the proposed role of radial F-actin organization during early neuronal development. However, they also raise concerns that need to be addressed in full before we can consider publication of the manuscript here.

Given these constructive comments, I would like to invite you to revise your manuscript with the understanding that the referee must be fully addressed and their suggestions taken on board. Please address all referee concerns in a complete point-by-point response. Acceptance of the manuscript will depend on a positive outcome of a second round of review. It is EMBO Reports policy to allow a single round of revision only and acceptance or rejection of the manuscript will therefore depend on the completeness of your responses included in the next, final version of the manuscript.

Supplementary/additional data: The Expanded View format, which will be displayed in the main HTML of the paper in a collapsible format, has replaced the Supplementary information. You can submit up to 5 images as Expanded View. Please follow the nomenclature Figure EV1, Figure EV2 etc. The figure legend for these should be included in the main manuscript document file in a section called Expanded View Figure Legends after the main Figure Legends section. Additional

Supplementary material should be supplied as a single pdf labeled Appendix. The Appendix includes a table of content on the first page with page numbers, all figures and their legends. Please follow the nomenclature Appendix Figure Sx throughout the text and also label the figures according to this nomenclature. For more details please refer to our guide to authors.

When preparing your letter of response to the referees' comments, please bear in mind that this will form part of the Review Process File, and will therefore be available online to the community. For more details on our Transparent Editorial Process, please visit our website:
http://emboj.embopress.org/about#Transparent_Process

Regarding data quantification, please ensure to specify the name of the statistical test used to generate error bars and P values, the number (n) of independent experiments underlying each data point (not replicate measures of one sample), and the test used to calculate p-values in each figure legend. Discussion of statistical methodology can be reported in the materials and methods section, but figure legends should contain a basic description of n, P and the test applied. Please also include scale bars in all microscopy images.

We now strongly encourage the publication of original source data with the aim of making primary data more accessible and transparent to the reader. The source data will be published in a separate source data file online along with the accepted manuscript and will be linked to the relevant figure. If you would like to use this opportunity, please submit the source data (for example scans of entire gels or blots, data points of graphs in an excel sheet, additional images, etc.) of your key experiments together with the revised manuscript. Please include size markers for scans of entire gels, label the scans with figure and panel number, and send one PDF file per figure.

- a complete author checklist, which you can download from our author guidelines (<http://emboj.embopress.org/authorguide#revision>). Please insert page numbers in the checklist to indicate where the requested information can be found.
 - a letter detailing your responses to the referee comments in Word format (.doc)
 - a Microsoft Word file (.doc) of the revised manuscript text
 - editable TIFF or EPS-formatted figure files in high resolution
- (In order to avoid delays later in the publication process please check our figure guidelines before preparing the figures for your manuscript:
http://www.embopress.org/sites/default/files/EMBOPress_Figure_Guidelines_061115.pdf)
- a separate PDF file of any Supplementary information (in its final format)
 - all corresponding authors are required to provide an ORCID ID for their name. Please find instructions on how to link your ORCID ID to your account in our manuscript tracking system in our Author guidelines (<http://emboj.embopress.org/authorguide>).

As part of the EMBO publication's Transparent Editorial Process, EMBO reports publishes online a Review Process File to accompany accepted manuscripts. This File will be published in conjunction with your paper and will include the referee reports, your point-by-point response and all pertinent correspondence relating to the manuscript.

I look forward to seeing a revised version of your manuscript when it is ready. Please let me know if you have questions or comments regarding the revision.

REFeree REPORTS

Referee #1:

This manuscript by Meka and Scharrenberg et al. demonstrates three major points - 1) Presence of centrosomal actin puncta and radiating filaments in the soma of young neurons, 2) Egress of the (presumably centrosomal) F-actin from the soma to neurites, and 3) requirement of the centrosomal protein PCM1 in organizing the somatic actin puncta. Overall, the concept is interesting, the writing is clear, and the data presentation is very nice. However there is a major technical concern that needs to be addressed, along with the following points:

1. Photobleaching must be accounted for in these experiments, and all the numerical data corrected for bleaching. This is very important to do anyway, but particularly in these studies as the differences are small and data ambiguous in some cases (see below). Please do the following: Take cells (such as the one in Fig. 2A), transfect with the respective photoconvertible construct (that was used in the actual experiments), then fix the sample with PFA BEFORE photoactivation. Now photoactivate the fixed cell and image using identical conditions as the real experiments (replace fixation buffer with imaging media before photoactivation). Quantify data from several cells. Since the fluorophores are fixed, the fall of fluorescence in this dataset is the photobleaching curve, please use this to normalize all datasets (PFA does not affect photoactivation of PAGFP, the others also should not be affected - in any case this will be clear once the experiment is done).

2. Numerous puncta are seen in the soma (Fig. 1D), but the authors seem to suggest that everything is related to centrosomal puncta. This does not make sense. Though there is an enrichment of puncta around centrosomes, this does not mean that the entire phenotype can be depicted as "centrosomal actin". Unless the authors can argue strongly as to why my point of view is incorrect, they need to revise their conclusions.

3. The radiating asters are not described well, or even or depicted clearly. What is the elongation rate etc.? Neither Fig. 1E (nor the movie) convincingly shows the elongating filaments. The authors should find a clearer way to demonstrate this point to the reader.

4. It is difficult to see the tip-enrichment in Fig. 2A. Scaling, insets etc. are needed to clarify this. Same with 2C, nothing is visible in the right-most image. Images of soluble (untagged) Eos 3.2 should be shown here as well, so the reader can compare the two datasets visually.

5. Cell in 3C is the same as 2A (same data is shown twice).

6. In Fig. 4A, where is the fluorescence going, if it is not getting into the soma? This effect may be entirely due to photobleaching. As mentioned previously, bleaching is a major concern in these experiments, and it is essential that the authors account for that. Actually, in these particular experiments, the authors need to make sure that there is no bleaching, so we can understand the kinetics of fluorescence.

7. It is important that the initial fluorescence-levels (absolute AFU levels) immediately after photoactivation are similar with each of the different probes used in this study. This is important, to be able to compare the differences in radial dispersion between the various photoactivatable probes used (for instance, a lack of sufficient targeting could simply be because much fewer molecules are activated in that case). Please add this data in Fig. 2, so the reader can judge. If the post-activation fluorescence levels are very different, the experiments should be repeated to ensure that the levels are comparable and the datasets are robust.

8. Same concern as 6) for Fig. 6C. The fluorescence is clearly decreasing after activation. Where are the activated molecules going, if not to the neuritic terminals?

9. Does SMIFH2 also block the radial movement of actin filaments? In the absence of any specific way to disrupt actin puncta, this becomes an important experiment to reinforce the co-relation of F-actin puncta and the radial dispersion into growth cones.

10. What is the relevance of the stable vs transient filaments on the radial movement of actin described in the paper? CofilinS3A experiments should be extended with PA-UtrGFP.

11. Actin distribution from soma to axon tips should be examined after CALI centrin2-KR. Authors explored Actin treadmilling in growth cone and total actin intensity in growth cones after CALI- Does the number of puncta of actin in the cell body also goes down? This will be an important piece of data to tie the centrosome and actin puncta nearby.

12. What is the effect of PCM1 shRNA on radial movement of actin into growth cones?

Other points:

- Some explanation is needed for the increased axonal lengths in Fig. 7
- In the discussion, the authors should expand a bit more on how their data relates to REFS #39, 40, and also the following references: Pfender et al., *Curr Biol.* 21:955-960 (2011); Schuh, 2011 *Nat. Cell Biol.*, 13:1431-1436.
- A better explanation of the rose plots - and how the reader needs to interpret them - is needed.
- "Irradiate" is not the best word to use, as the connotation is that it is something bad. How about 'activate'?"
- PCM-1: Acronym should be defined where it first appears, and also please add a brief description of what the protein does
- The association of PCM-1 with actin (post drug-treatment) is very interesting. The authors may want to speculate a bit in the discussion, what this might mean.
- The title would be more impactful if it was more specific.

Referee #2:

In this manuscript, Meka et al. describe the organization of actin in the cell body of immature neurons and its role for neuronal development. They found actin puncta in proximity of the centrosome, that nucleate radial filaments. This is the source of a centrifuge movement of actin that is linked to the presence of filamentous actin at the extremities of neuronal processes.

Overall, this is a thorough study with an impressive amount of work, with quality data mostly obtained by a range of imaging techniques. I think this is an important work that unravels a novel actin organization in neurons and its potential role for proper neuronal development. As stated in the manuscript, the mechanistic details of this process are not yet known from this study, but this is a good starting point and its publication would be of great interest for the neuronal cell biology community.

Before recommending acceptance in EMBO Reports, I have a number of points that I would like the authors to address in a revised version of the manuscript.

Major points

1- How this new actin movement takes place within the known organization of actin in immature neurons is not very clear. In the Introduction, it would be good to present in a clearer way the classical view of microtubules pushing and actin pulling to regulate neurite growth, as initially hypothesized by Forscher and Smith (*JCB* 1988, reviewed in Schelski & Bradke *MCN* 2017); as well as contradictory views on the role of actin (Chia *MBoC* 2017). In particular, the phrase "impaired local assembly is sufficient to block neurite growth" seems at odds with experiments showing that disassembly of actin (by drugs, cofilin etc) has a positive effect on neurite growth in a number of studies - as also shown in the present manuscript for cytochalasin D treatment (Supp. Fig. 12). It would also be interesting to discuss how the centrifugal movement described here integrates with the more local-scale retrograde actin flow that occurs within processes. In this view, the centrifugal movement of actin from the cell body to neurites supplies filamentous actin that is able to restrict neurite growth, as demonstrated by the longer neurites after PCM1 knockdown (Fig. 7I-J). Overall, does the radial movement of actin from the cell body have a positive or negative effect on neurite growth?

2- Related to this, it would be interesting to assess the downstream effect of different perturbations of the radial actin flow on neurite growth in cultured neurons, similar to what is done for PCM1 knockdown in Supp. Fig. 12. In particular, what is the longer-term effect of Cofilin S3E and Drebrin S142D on neurite growth? What about the effect of Centrin2 CALI? Would the stabilization of actin and reduced threadmilling have a negative effect on growth (i.e. reverse effect of cytochalasin

destabilization) or would the reduced movement promote growth?

Minor points

- 3- Is it possible that the quadrant-based measurements (for actin puncta in Fig. 1B, Supp. Fig. 1B, Supp. Fig. 15, and EB3 comets in Supp. Fig. 4B) are biased by the presence of the nucleus? From the image in Fig. 1A and Supp Fig. 4A, it seems that the centrosome sits in the soma region where most of the the non-nuclear material is present, so I would expect a significant asymmetry for any non-nuclear structure in this region. How to take this into account to detect a meaningful concentration of actin puncta? From the material and methods, it seems that the nucleus is treated as part of the soma volume for the analysis. Would a normalization to the non-nuclear volume help?
- 4- The description of the photo-activation experiments and their interpretation seem to overlook the fact that LifeAct and UtrCH will constantly exchange on actin filaments over the time of the experiment. So, I'm not sure how they can be used as a proof that filamentous actin is being transported - could the author clarify this point? It is interesting that actin itself as a different movement with both centrifuge movement from the cell body and retrograde movement from the processes - Could it be made clearer how this does this with the authors' model?
- 5- On the SMLM images of Supp. Fig. 2, it is not clear where the centrioles are in the zoomed areas (1, 2, 3). Could an overlay with the non-SR pericentrin image or dashes to indicate their positions in the zoomed imaged be used to clarify the actin distribution relative to their approximative localization?
- 6- It would be useful to explain what the z-score is when introducing measurements of effects on actin threadmilling and puncta intensity (first in Fig. 6B), as this is not an intuitive metric.

Referee #3:

In this very interesting article the authors describe in great detail the characteristics of actin puncta in the vicinity of the centrosome in the soma. This in itself is an interesting novel observation. In addition, the pericentriolar protein PCM-1 is shown to be required for the formation of actin puncta and importantly neuronal differentiation, I agree with the authors statement that this is an interesting new finding.

In general, the manuscript does not integrate previously published findings of actin on endo-membranes (see Alekhina et al. *J Cell Sci.* 2017 Jul 15;130(14):2235-224) enough. The area around the centrosome is often populated with internal membranes like endosomes (WASH), Golgi and ER (WHAMM, JMY) that are known to be associated with actin patches and polymerisation factors (in brackets). The authors need to explore endo-membranes as possible source of their actin puncta more extensively to exclude the possibility that their inhibitor and siRNA treatments affect endosomal and secretory compartment membranes that could be the underlying cause of the changes in neuronal development.

I do not feel able to comment on the biological importance of radial actin structures in neurons. Similarly, why it is important that it is F-actin that is distributed but not G-actin is not immediately obvious to me. This will always be difficult to ascertain given the dynamic turnover of actin monomers in F-actin filaments. Please see my comments below for details.

Major:

- I am really sorry and I might have missed it, but where are the materials and methods? If they are not included, the manuscript is too long for the format. I think there are a few redundant experiments in this report, seven figures and sixteen supplementary figures make it hard to focus on the key findings.

- Quantification methods are quite inconsistent between experiments. To help the reader I think it would be useful to use the same quantification for each treatment where appropriate. This could include: F-actin puncta number (similar to Fig 5E), F-actin average distance to centrosome (possibly simplify Fig 1B?); F-actin puncta size and movement.

- Quantification analysis details are unclear and need to be explained in methods. For example , how are the intensity and ratios measured for the photoconversion/activation? At what region in the growth cone is this measured, and how was the growth cone traced? Also, at what time point on the graph is the 0 sec - is it the point of activation/conversion?

- All actin images need to be co-stained with centrosomal markers or the position of centrosome

needs to be pointed out (for example in Figs. 1E; S2A SMLM inset 3) to make interpretation of the images possible.

-Fig.2: the indicated area of photoactivation/conversion is very large, and in my opinion does not preclude activation/conversion of a substantial pool of actin monomers at the same time as F-actin structures. I do not think the questions of filamentous versus monomeric actin will be able to be solved without doubt and the authors should not try to solve it in this manuscript.

-Fig. S8: Cytochalasin D, like the authors state, does disrupt F-Actin and therefore no new F-actin structures should be formed in periphery or soma. The failure to detect lifeact bound actin mediated fluorescence increase in periphery could also be due to dilution of signal in during diffusion. The authors need to show that, for example, diffusion of photoactivatable GFP would lead to measurable increase in fluorescence.

-PCM-1 data, I agree with the conclusion that lack of PCM-1 also seems to affect more distant areas of the cell. However, not I am not sure I would call it radial (see also comments on F-actin vs. G-actin distribution).

Minor:

-PCM-1 is not Fig. 5 as mentioned in text, but Fig.7.

- Authors should include details for incubation times for chemicals. For example, the duration of SiR-Actin-, cytochalasin D-, Brefeldin-A-, Jasplakinolide- and Nocodazole treatment are not mentioned.

1st Revision - authors' response

8 July 2019

Referee #1:

This manuscript by Meka and Scharrenberg et al. demonstrates three major points - 1) Presence of centrosomal actin puncta and radiating filaments in the soma of young neurons, 2) Egress of the (presumably centrosomal) F-actin from the soma to neurites, and 3) requirement of the centrosomal protein PCM1 in organizing the somatic actin puncta. Overall, the concept is interesting, the writing is clear, and the data presentation is very nice. However, there is a major technical concern that needs to be addressed, along with the following points:

1. Photobleaching must be accounted for in these experiments, and all the numerical data corrected for bleaching. This is very important to do anyway, but particularly in these studies as the differences are small and data ambiguous in some cases (see below). Please do the following: Take cells (such as the one in Fig. 2A), transfect with the respective photoconvertible construct (that was used in the actual experiments), then fix the sample with PFA BEFORE photoactivation. Now photoactivate the fixed cell and image using identical conditions as the real experiments (replace fixation buffer with imaging media before photoactivation). Quantify data from several cells. Since the fluorophores are fixed, the fall of fluorescence in this dataset is the photobleaching curve, please use this to normalize all datasets (PFA does not affect photoactivation of PaGFP, the others also should not be affected - in any case this will be clear once the experiment is done).

We would like to thank this Reviewer for her/his positive comments on our work. We agree with this Reviewer regarding the technical concern of photobleaching. Accordingly, we have performed the suggested control experiments (New Supplementary Figure 7). Overall, the PaGFP-UtrCH probe is performing well without considerably photobleaching after photoactivation in the fixed cells (New Supplementary Figure 7J). On the other hand, PaGFP alone is photobleaching around 50% after 120 sec imaging. We consider that this is not an issue for our analysis given that we used this probe to test the neurite-specific distribution of PaGFP-UtrCH compared to the PaGFP alone (New Figure 2E). In other words, our comparison is qualitative rather than quantitative. Moreover, the initial fluorescence-levels immediately after photoactivation for PaGFP are considerably higher than the PaGFP compared with the PaGFP-UtrCH (New Supplementary Figure 7A). In fact, we could not detect drastic differences of photobleaching in our time-lapses of living cells with PaGFP alone (e.g. Video 4). This suggest that the signal of PaGFP alone could be affected by PFA fixation.

Regarding the photobleaching of Eos probes, we could not induce photoconversion of the probes in fixed samples with the imaging parameters used in living samples. We managed to photoconvert all the Eos probes, however, modifying our original imaging parameters of living cells (Material and Methods, *Photoconversion experiments*). Therefore, we consider that with the values obtained in fixed samples it is not possible to adjust our values from living samples. We added all the information in the New Supplementary Figure 7 so the readers can judge by themselves. Importantly, the data obtained with Eos probes is qualitatively similar to the experiments performed with PaGFP-UtrCH (see New Figure 2E and New Supplementary Figure 8F). Thus, we decided to perform all the new experiments with PaGFP-UtrCH and repeated some experiments (previously done with Lifeact-mEos3.2) with PaGFP-UtrCH. All these experiments are included in the revised version of the manuscript.

2. Numerous puncta are seen in the soma (Fig. 1D), but the authors seem to suggest that everything is related to centrosomal puncta. This does not make sense. Though there is an enrichment of puncta around centrosomes, this does not mean that the entire phenotype can be depicted as "centrosomal actin". Unless the authors can argue strongly as to why my point of view is incorrect, they need to revise their conclusions.

This Referee is right, and we have several F-actin puncta in the cell body and not all are necessarily associated with centrosomal functions. The image from Fig. 1D, however, is a max projection of cytosolic and cortical F-actin puncta. Therefore, we also included insets of image stacks taken at the focal plane of the centrosome. Importantly, our analysis of Centrin2 CALI shows that a proportion of these somatic F-actin puncta are lost after the transient centrosome inactivation (New Figure 5B, C). Thus, we could conclude that a proportion of somatic F-actin puncta are associated with the centrosome, not only physically but functionally as well.

3. The radiating asters are not described well, or even or depicted clearly. What is the elongation rate etc.? Neither Fig. 1E (nor the movie) convincingly shows the elongating filaments. The authors should find a clearer way to demonstrate this point to the reader.

In the revised version of our manuscript we included new analysis of the extension frequency of the F-actin asters (New Figure 1F). Moreover, we selected better examples to demonstrate the behavior of the radiating asters (New Figure 1E).

4. It is difficult to see the tip-enrichment in Fig. 2A. Scaling, insets etc. are needed to clarify this. Same with 2C, nothing is visible in the right-most image. Images of soluble (untagged) Eos 3.2 should be shown here as well, so the reader can compare the two datasets visually.

We changed the figures according to the Referee comments. In the revised version of the manuscript we included representative neurite tips in the New Figures 2D, 3B, 6K, 7E, and Supplementary Figure 8E.

5. Cell in 3C is the same as 2A (same data is shown twice).

We included another example (New Supplementary Figure 8A) to avoid having the same data twice.

6. In Fig. 4A, where is the fluorescence going, if it is not getting into the soma? This effect may be entirely due to photobleaching. As mentioned previously, bleaching is a major concern in these experiments, and it is essential that the authors account for that. Actually, in these particular experiments, the authors need to make sure that there is no bleaching, so we can understand the kinetics of fluorescence.

Given that the growth cone has a cycle of F-actin polymerization/depolymerization, it is possible that the signal of Lifeact-Eos 3.2 or PaGFP-UtrCH decreases overtime independently of photobleaching. Moreover, our new control experiments aim to determine the degree of photobleaching of our probes show that PaGFP-UtrCH is not bleaching under our imaging settings (New Supplementary Figure 7). Furthermore, overexpressing CofilinS3E mutant, which stabilizes F-actin, the intensity of the converted signal, which reach the growth cone, is

higher when the F-actin is stabilized (New Supplementary Figure 13A-C). Most likely, this effect is due to decreased F-actin turnover in the growth cones after F-actin stabilization.

7. It is important that the initial fluorescence-levels (absolute AFU levels) immediately after photoactivation are similar with each of the different probes used in this study. This is important, to be able to compare the differences in radial dispersion between the various photoactivatable probes used (for instance, a lack of sufficient targeting could simply be because much fewer molecules are activated in that case). Please add this data in Fig. 2, so the reader can judge. If the post-activation fluorescence levels are very different, the experiments should be repeated to ensure that the levels are comparable and the datasets are robust.

The initial fluorescence-levels immediately after photoactivation/photoconversion now are included in the revised version of the manuscript (New Supplementary Figure 7). We plotted all cells used for each condition tested. We found that the major differences in the initial fluorescence-levels after photoactivation/conversion are when using the PaGFP and Eos alone, which generally are high compared to PaGFP-UtrCH or Lifeact-Eos, respectively (New Supplementary Figure 7A, G). Additionally, when Cytochalasin D was used together with PaGFP or PaGFP-UtrCH we obtained high values of activated signal, although the values are not significantly different from the other conditions tested in that particular experiment (New Supplementary Figure 7B). When the probes are attached to Lifeact or UtrCH, however, the initial fluorescence-levels immediately after photoactivation / photoconversion are quite consistent for each independent experiment.

We noticed that it is difficult to achieve similar initial fluorescence-levels across all the experiments using PaGFP-UtrCH given that we only see the level of expression once the probe is photoactivated. With Lifeact-Eos the initial signal (before photoconversion) was a good indication for the posterior levels of photoconverted signal (New Supplementary Figure 7H). We repeated our experiments several times trying to achieve that with PaGFP-UtrCH using our reporters (Maroon or tDimer) as an indirect indication of expression levels of PaGFP-UtrCH. We were able to get similar levels of initial photoactivated signal for independent experiments (see New Supplementary Figure 7B-F). Therefore, for individual experiments our controls and experimental conditions properly match. Importantly, even with low-quantity or high-quantity of photoactivated signal we detected the same behavior of radial spread independently of using Lifeact-Eos or PaGFP-UtrCH (see New Figure 2E and Supplementary Figure 8F). Basically, we are describing our data in a more qualitative, rather than in a quantitative manner.

We decided to incorporate these data in the New Supplementary Figure 7 given the lack of space in the main Figures.

8. Same concern as 6) for Fig. 6C. The fluorescence is clearly decreasing after activation. Where are the activated molecules going, if not to the neuritic terminals?

We repeated these experiments using PaGFP-UtrCH given that this probe is not photobleaching under our imaging settings (New Figure 5F, G; New Supplementary Figure 9D, E; New Supplementary Figure 12C, D).

9. Does SMIFH2 also block the radial movement of actin filaments? In the absence of any specific way to disrupt actin puncta, this becomes an important experiment to reinforce the co-relation of F-actin puncta and the radial dispersion into growth cones.

Using SMIFH2 we detected that radial movement of F-actin is reduced considerably (New Figure 7D-F). Thus, suggesting that somatic F-actin puncta is relevant for radial translocation into growth cones.

10. What is the relevance of the stable vs transient filaments on the radial movement of actin described in the paper? CofilinS3A experiments should be extended with PA-UtrGFP.

We believe that the Referee meant to extend our experiments of CofilinS3E which stabilize F-actin and not CofilinS3A (that was not used in this study). In the revised version of the manuscript we included CofilinS3E mutant together with PaGFP-UtrCH. We found that the movement of the activated signal is slower when the F-actin is stabilized. However, the

intensity of the converted signal, which reaches the growth cone, is higher when the F-actin is stabilized (New Supplementary Figure 13A-C). Most likely this effect is due to decreased F-actin turnover in the growth cones after F-actin stabilization.

11. Actin distribution from soma to axon tips should be examined after CALI centrin2-KR. Authors explored Actin treadmilling in growth cone and total actin intensity in growth cones after CALI- Does the number of puncta of actin in the cell body also goes down? This will be an important piece of data to tie the centrosome and actin puncta nearby.

In the revised version of our manuscript we extended our analysis and found that somatic F-actin puncta are reduced after CALI (New Figure 5B, C).

12. What is the effect of PCM1 shRNA on radial movement of actin into growth cones?

In the revised version of the manuscript we included PCM-1 shRNA together with PaGFP-UtrCH. We found that the intensity of the converted signal, which reaches the neurite tip, is lower when PCM-1 is down-regulated (New Figure 6J-L).

Other points:

- Some explanation is needed for the increased axonal lengths in Fig. 7.

We consider that F-actin at growth cones have an inhibitory role over neurite growth. Thus, as suggested by this Reviewer, we now included experiments where we tested the effect of PCM-1 down-regulation on the centrifugal movement of PaGFP-UtrCH. We found that the intensity of the converted signal, which reach the neurite tip, is lower when PCM-1 is down-regulated (New Figure 6J-L). Thus, supporting our initial results where we described that after PCM-1 down-regulation there is less F-actin in the neurite tips, which ultimately promotes neurite overgrowth (New Figure 6C, F). Moreover, we now include a Figure for Reviewers (Figure 1a for Reviewers) where we tested the effect of F-actin on neurite growth. We disrupted F-actin polymerization using Cytochalasin D; thus, promoting neurite overgrowth. The drug was removed and F-actin reorganization/repolymerization at the neurite tip induced a reduction of the previously induced neurite overgrowth. Basically, neurites retracted once the F-actin at the growth cones is reorganized. This suggests that F-actin at growth cones has a negative effect on neurite growth (more F-actin = less neurite growth). Accordingly, the distribution of the activated signal in the photo activation/conversion paradigms in control cells is enriched preferentially in the shorter neurites (New Figure 2E and Supplementary Figure 8F).

- In the discussion, the authors should expand a bit more on how their data relates to REFS #39, 40, and also the following references: Pfender et al., *Curr Biol.* 21:955-960 (2011); Schuh, 2011 *Nat. Cell Biol.*, 13:1431-1436.

In the new version of the discussion we considered the information provided by the manuscript mentioned above.

- A better explanation of the rose plots - and how the reader needs to interpret them - is needed.

In the revised version of the manuscript, we now included the F-actin puncta distribution using the centrosome position as a reference. For this we considered cytosolic puncta only. First, we removed the nuclear volume. All cytosolic puncta were considered if there was no nuclear volume between them and the centrosome. The remaining space was normalized to plot the relative distribution of the F-actin puncta regarding the centrosome position. Additionally, we plot the absolute distance of the F-actin puncta to the centrosome and show the real 3D distribution. We consider that this new representation of our data is easier to interpret.

- "Irradiate" is not the best word to use, as the connotation is that it is something bad. How about 'activate'?

We now changed irradiate for illumination/activation according to this Reviewer comments.

- PCM-1: Acronym should be defined where it first appears, and also please add a brief description of what the protein does.

We now included the definition of PCM-1 where it first appears (Abstract).

- The association of PCM-1 with actin (post drug-treatment) is very interesting. The authors may want to speculate a bit in the discussion, what this might mean.

In the revised version of our manuscript, we included a discussion regarding the spatial relation between PCM-1/F-actin evident after microtubules disruption.

- The title would be more impactful if it was more specific.

The new title is: “Radial somatic F-actin organization affects growth cone dynamics during early neuronal development”

Referee #2:

In this manuscript, Meka et al. describe the organization of actin in the cell body of immature neurons and its role for neuronal development. They found actin puncta in proximity of the centrosome, that nucleate radial filaments. This is the source of a centrifuge movement of actin that is linked to the presence of filamentous actin at the extremities of neuronal processes. Overall, this is a thorough study with an impressive amount of work, with quality data mostly obtained by a range of imaging techniques. I think this is an important work that unravels a novel actin organization in neurons and its potential role for proper neuronal development. As stated in the manuscript, the mechanistic details of this process are not yet known from this study, but this is a good starting point and its publication would be of great interest for the neuronal cell biology community.

Before recommending acceptance in EMBO Reports, I have a number of points that I would like the authors to address in a revised version of the manuscript.

Major points.

1- How this new actin movement takes place within the known organization of actin in immature neurons is not very clear. In the Introduction, it would be good to present in a clearer way the classical view of microtubules pushing and actin pulling to regulate neurite growth, as initially hypothesized by Forscher and Smith (JCB 1988, reviewed in Schelski & Bradke MCN 2017); as well as contradictory views on the role of actin (Chia MBoC 2017). In particular, the phrase "impaired local assembly is sufficient to block neurite growth" seems at odds with experiments showing that disassembly of actin (by drugs, cofilin etc) has a positive effect on neurite growth in a number of studies - as also shown in the present manuscript for cytochalasin D treatment (Supp. Fig. 12).

We would like to thank this Reviewer for her/his positive comments on our work. In the revised version of our manuscript we changed the introduction according to this Reviewer comments (first paragraph of the introduction). The phrase is now as follows: “For instance, numerous studies have demonstrated that F-actin is assembled locally in growth cones and that impaired local assembly is sufficient to affect neurite growth. Moreover, we include the citation from Chia, J. X. et al. MBoC, 2017.

It would also be interesting to discuss how the centrifugal movement described here integrates with the more local-scale retrograde actin flow that occurs within processes. In this view, the centrifugal movement of actin from the cell body to neurites supplies filamentous actin that is able to restrict neurite growth, as demonstrated by the longer neurites after PCM1 knockdown (Fig. 7I-J). Overall, does the radial movement of actin from the cell body have a positive or negative effect on neurite growth?

We now included experiments where we tested the effect of PCM-1 down-regulation on the centrifugal movement of PaGFP-UtrCH. We found that the intensity of the converted signal,

which reaches the neurite tip, is lower when PCM-1 is down-regulated (New Figure 6J, L). Thus, supporting our initial results, where we described that after PCM-1 down-regulation there is less F-actin in the neurite tips. Moreover, we now include a Figure for Reviewers (Figure 1a for Reviewers) where we tested the effect of F-actin on neurite growth. We disrupted F-actin polymerization using Cytochalasin D; thus, promoting neurite overgrowth. The drug was removed and F-actin reorganization/repolymerization at the neurite tip induced a reduction of the previously induced neurite overgrowth. Basically, neurites retracted once the F-actin at growth cones is reorganized. This suggests that F-actin at growth cones has a negative effect on neurite growth (more F-actin = less neurite growth). Accordingly, the distribution of the activated signal in the photoactivation/conversion paradigms in control cells is enriched preferentially in the shorter neurites (New Figure 2E and Supplementary Figure 8F).

2- Related to this, it would be interesting to assess the downstream effect of different perturbations of the radial actin flow on neurite growth in cultured neurons, similar to what is done for PCM1 knockdown in Supp. Fig. 12. In particular, what is the longer-term effect of Cofilin S3E and Drebrin S142D on neurite growth? What about the effect of Centrin2 CALI? Would the stabilization of actin and reduced treadmilling have a negative effect on growth (i.e. reverse effect of cytochalasin destabilization) or would the reduced movement promote growth?

Stabilization of F-actin with the expression of Cofilin S3E leads to a reduced neurite length compared to cells expressing Cofilin WT (New Supplementary Figure 13 D, E). Cofilin WT over-expression promoted elongation of the longest neurite, compared with control cells (New Supplementary Figure 13 D, E), This neurite elongation, however, is reversed to the control-like situation when Cofilin S3E construct was overexpressed. Of note, we did not detect differences in neurite elongation between Drebrin S142D and Drebrin WT overexpression. This is possibly due to the reported complex interaction between Drebrin, F-actin, and microtubules (Worth D. C. et al J. Cell Biol. 2013). In fact, Drebrin S142D not only induced F-actin bundles (reducing F-actin dynamics) but also promotes the insertion of microtubules into growth cone filopodia (Worth D. C. et al J. Cell Biol. 2013).

The effect of Centrin2 CALI should be transient since the expression of Centrin2-KillerRed is constant and the illuminated Centrin2 might be exchanged over time. Thus, restoring centrosomal functions and precluding the possibility to test the long-term effect of Centrin2 CALI.

Given our data, we consider that the reduced treadmilling might not be an absolute parameter reliable for all conditions to compare neurite growth. Reduced treadmilling obtained after F-actin stabilization using Cofilin or Drebrin mutants and after PCM-1 down-regulation or Centrin2 CALI could arise due to different reasons. In the first condition (Cofilin and Drebrin mutants), F-actin content was not depleted at the growth cones. However, after PCM-1 down-regulation, cytochalasin D treatment or Centrin2 CALI, F-actin content in growth cones/neurite tips is reduced significantly. Thus, affecting the F-actin treadmilling negatively. To test this directly, we treated cells with Cytochalasin D, depleting F-actin content in neurites tips. After this treatment, we found that neurite tips in general preserve an accumulation of F-actin (as reported by Chia, J. X. et al. MBoc, 2017) with a reduced treadmilling (Figure 1C for Reviewers). Neurites, however, still elongated. Cytochalasin D washout, induced F-actin reorganization in the neurite tips with the consequent increment on F-actin treadmilling and neurite retraction.

Overall, we could conclude that F-actin content reduction is associated with reduced F-actin treadmilling and neurite elongation. However, reduced F-actin treadmilling without changing F-actin content is not necessarily associated with inhibition of neurite growth.

Minor points.

3- Is it possible that the quadrant-based measurements (for actin puncta in Fig. 1B, Supp. Fig. 1B, Supp. Fig. 15, and EB3 comets in Supp. Fig. 4B) are biased by the presence of the nucleus? From the image in Fig. 1A and Supp Fig. 4A, it seems that the centrosome sits in the soma region where most of the the non-nuclear material is present, so I would expect a significant asymmetry for any non-nuclear structure in this region. How to take this into account to detect a meaningful concentration of actin puncta? From the material and methods, it seems that the nucleus is treated as part of the soma volume for the analysis. Would a normalization to the non-nuclear volume help?

In the revised version of the manuscript, we now consider the volume occupied by the nucleus. We included the F-actin puncta distribution using the centrosome position as a reference. For this we considered cytosolic puncta only. First, we removed the nuclear volume. All cytosolic puncta were considered if there was no nuclear volume between them and the centrosome. The remaining space was normalized to plot the relative distribution of the F-actin puncta regarding the centrosome position. Additionally, we plot the absolute distance of the F-actin puncta to the centrosome and show a 3D representation of the actual distances.

4- The description of the photo-activation experiments and their interpretation seem to overlook the fact that LifeAct and UtrCH will constantly exchange on actin filaments over the time of the experiment. So, I'm not sure how they can be used as a proof that filamentous actin is being transported - could the author clarify this point? It is interesting that actin itself as a different movement with both centrifuge movement from the cell body and retrograde movement from the processes - Could it be made clearer how this does this with the authors' model?

We agree with this Reviewer and it is not easy to discriminate if the probes used in this study are “jumping” from one actin filament to another or if they are showing directly F-actin translocation.

We consider that we have evidence supporting the actin filaments translocation. First, we could not detect a potential “jumping” effect when the Lifeact and UtrCH probes were photoactivated in growth cones (New Supplementary Figure 10A, 11A). The converted signal was restricted to the growth cones and we could not detect retrograde converted signal moving out from the growth cones. Although, it is possible that the “jumping” signal is inserted into the growth cone F-actin treadmilling. However, if that is the case, then we should call into question analyses of F-actin treadmilling in growth cones using these probes. Second, and as the Reviewer stated, photoconversion of actin alone has a different behavior than Lifeact. A fraction of converted actin signal in the growth cone translocated retrogradely (presumably the monomers). Finally, we expressed Drebrin mutant (DrebrinS142D), which promotes and binds F-actin bundles. We could detect not only the Lifeact but also the DrebrinS142D translocation radially in a comet-like fashion (New Figure 4B). Certainly, it could be that DrebrinS142D is also following a “jumping” effect. However, the same argument could apply while tracing EB3 signal in growing microtubules and nowadays EB3 movement is a reliable approach to trace growing microtubules.

In summary, we agree with this Reviewer concern but we believe our data and experimental approaches might open new avenues to better understand F-actin dynamics in growing neurons.

5- On the SMLM images of Supp. Fig. 2, it is not clear where the centrioles are in the zoomed areas (1, 2, 3). Could an overlay with the non-SR pericentrin image or dashes to indicate their positions in the zoomed imaged be used to clarify the actin distribution relative to their approximative localization?

In the revised version of the manuscript we now included the centrioles position of the SMLM images.

6- It would be useful to explain what the z-score is when introducing measurements of effects on actin treadmilling and puncta intensity (first in Fig. 6B), as this is not an intuitive metric.

We would like to apologize for not having been clear enough regarding our analysis and the presentation of our data. In the revised version of the manuscript, this is now clarified in the Material & Methods (*Statistical analysis*). Z-score normalization was done for data sets with multiple populations (e.g. distance of puncta to centrosome for each cell) by centering values around the mean and dividing by the standard deviation. Resulting values are therefore expressed as standard deviations (σ) from the mean.

Referee #3:

In this very interesting article the authors describe in great detail the characteristics of actin puncta in the vicinity of the centrosome in the soma. This in itself is an interesting novel observation. In addition, the pericentriolar protein PCM-1 is shown to be required for the formation of actin puncta

and importantly neuronal differentiation, I agree with the authors statement that this is an interesting new finding.

In general, the manuscript does not integrate previously published findings of actin on endo-membranes (see Alekhina et al. J Cell Sci. 2017 Jul 15;130(14):2235-224) enough. The area around the centrosome is often populated with internal membranes like endosomes (WASH), Golgi and ER (WHAMM, JMY) that are known to be associated with actin patches and polymerisation factors (in brackets). The authors need to explore endo-membranes as possible source of their actin puncta more extensively to exclude the possibility that their inhibitor and siRNA treatments affect endosomal and secretory compartment membranes that could be the underlying cause of the changes in neuronal development.

We would like to thank this Reviewer for her/his positive comments on our work. We agree with this Reviewer regarding the potential relation between endo-membranes and somatic F-actin puncta. Therefore, in the revised version of our manuscript we included experiments testing the effect of Brefeldin A (BFA), which disrupts endo-membranes (Golgi and ER), on somatic F-actin puncta. We found that the number of somatic F-actin puncta is not affected after BFA treatment (New Supplementary Figure 18).

I do not feel able to comment on the biological importance of radial actin structures in neurons. Similarly, why it is important that it is F-actin that is distributed but not G-actin is not immediately obvious to me. This will always be difficult to ascertain given the dynamic turnover of actin monomers in F-actin filaments. Please see my comments below for details.

Major:

- I am really sorry and I might have missed it, but where are the materials and methods? If they are not included, the manuscript is too long for the format. I think there are a few redundant experiments in this report, seven figures and sixteen supplementary figures make it hard to focus on the key findings.

We would like to apologize for this confusion. In the revised manuscript we made our materials and methods more explicit. In addition, we did our best to better present our data to show our key findings.

- Quantification methods are quite inconsistent between experiments. To help the reader I think it would be useful to use the same quantification for each treatment where appropriate. This could include: F-actin puncta number (similar to Fig 5E), F-actin average distance to centrosome (possibly simplify Fig 1B?); F-actin puncta size and movement.

We changed some quantification methods (e.g. rose plots in New Figures 1B, 7C, Supplementary Figure 1B, and 18C) to make a better representation of our data. We tried to keep the same quantification method for all experiments. However, not always it is possible given the different outcome for each experiment setup. For instance, we cannot quantify F-actin puncta in time-lapse experiments (2 dimensions over time; Supplementary Figure 4B) as we did for somatic F-actin puncta in fixed samples (3 dimensions; New Figures 1B, 7C, Supplementary Figure 1B, and 18C).

- Quantification analysis details are unclear and need to be explained in methods. For example, how are the intensity and ratios measured for the photoconversion/activation? At what region in the growth cone is this measured, and how was the growth cone traced? Also, at what time point on the graph is the 0 sec - is it the point of activation/conversion?

We would like to apologize for this issue. In the revised version of our manuscript we improve the description of our analysis (Material & Methods; *Photoconversion / Photoactivation analysis*).

- All actin images need to be co-stained with centrosomal markers or the position of centrosome needs to be pointed out (for example in Figs. 1E; S2A SMLM inset 3) to make interpretation of the images possible.

We now included centrosomal markers in our images, where F-actin is shown (e.g. New Figure 1E and Supplementary Figure 2).

-Fig.2: the indicated area of photoactivation/conversion is very large, and in my opinion does not preclude activation/conversion of a substantial pool of actin monomers at the same time as F-actin structures. I do not think the questions of filamentous versus monomeric actin will be able to be solved without doubt and the authors should not try to solve it in this manuscript.

We activated single F-actin puncta at the soma, but there was not enough converted signal to perform a proper analysis (New Supplementary Figure 7K). Therefore, we decided to increase the illumination area.

We agree, however, with this Reviewer and we consider it is not easy to discriminate between labelling the actin filament alone or together with the actin monomers as well, with the probes used in this study. It was reported that Lifeact is binding both (monomers and polymers; Riedl, J. et al. *Nat Methods* 2008). However, to our knowledge UtrCH is more specifically binding the polymers (Melak, M. et al. *J Cell Sci* 2017; Burkel, B. M. et al. *Cell Motil Cytoskeleton* 2007). Importantly, we performed an experiment aimed to clarify the nature of the Lifeact converted signal that is translocated: F-actin was disrupted using Cytochalasin D before photo conversion / activation. If the probe is also binding monomers, we should detect a fraction of the signal still translocating. However, we could not detect translocation of converted signal suggesting that we are mainly tracking the movement of the polymer (New Supplementary Figure 9 D, E).

In summary, we agree with this Reviewer concern but we believe our data and experimental approaches might open new avenues to better understand F-actin dynamics in growing neurons.

-Fig. S8: Cytochalasin D, like the authors state, does disrupt F-Actin and therefore no new F-actin structures should be formed in periphery or soma. The failure to detect lifeact bound actin mediated fluorescence increase in periphery could also be due to dilution of signal in during diffusion. The authors need to show that, for example, diffusion of photoactivatable GFP would lead to measurable increase in fluorescence.

Yes, this is an important point we did not consider before. In the revised version of our manuscript we include this experiment. We found that Cytochalasin D treatment before photo activation of PaGFP alone did not preclude translocation of the activated GFP signal towards the cell periphery (New Supplementary Figure 9 D, E)

-PCM-1 data, I agree with the conclusion that lack of PCM-1 also seems to affect more distant areas of the cell. However, not I am not sure I would call it radial (see also comments on F-actin vs. G-actin distribution).

In the revised version of the manuscript we included PCM-1 shRNA together with PaGFP-UtrCH (which preferentially binds the polymer) photoactivation. We found that the intensity of the converted signal, which reaches the neurite tip, is lower when PCM-1 is down-regulated (New Figure 6J-L). Therefore, we consider that PCM-1 regulates the centrifugal/radial translocation of the activated UtrCH signal.

Minor:

-PCM-1 is not Fig. 5 as mentioned in text, but Fig.7.

We would like to apologize for this mistake, which has been corrected in our revised version of the manuscript.

- Authors should include details for incubation times for chemicals. For example, the duration of SiR-Actin-, cytochalasin D-, Brefeldin-A-, Jasplakinolide- and Nocodazole treatment are.

In our revised version of the manuscript, we clarified all the details in the Material & Methods. We apologize for this issue.

[Figures for referees not shown.]

2nd Editorial Decision

28 August 2019

Thank you for submitting the revised version of your manuscript. It has now been seen by two of the original referees.

As you can see, both referees find that the study is significantly improved during revision and recommend publication. Before I can accept the manuscript, I need you to address some editorial points below:

- Please address the remaining concerns of the referees. Referee #2 asks for a clearer explanation of the proposed model in the manuscript text and some textual corrections. Please address the minor concern of referee #3 textually - I can see that you already responded to this concern during the revision by providing additional controls, however, referee #2 does not seem fully convinced. Please add a sentence or two in the discussion acknowledging this shortcoming.

Thank you again for giving us to consider your manuscript for EMBO Reports, I look forward to your minor revision.

REFeree REPORTS**Referee #2:**

In this revised version, Meka et al. have addressed the points made in my review, in particular the experimental ones, with a clearer presentation of results. It's still not very clear to me from the manuscript that the authors favor a model where actin at the tip of neurites would have a negative effect on neurite growth and how experiments that examines neurite growth as an output fit in this model (as stated and explained in the response to reviewers), but this is a presentation issue that is quite minor. Other very minor points: the second to last phrase of the introduction's first paragraph ("Thus, adding weight...") seems to be missing a verb; and in the last phrase of the same paragraph ("To test this possibility, we performed..."), I guess the authors mean "experiments" rather than "methods". I think the manuscript is now suitable for publication in EMBO Reports, and look forward to this interesting addition to the literature on actin organization in neurons.

Referee #3:

The authors have now answered most of my queries and have improved the manuscript.

It is a bit curious that the authors chose only to use Brefeldin A to look at endomembranes as possible source of the actin puncta and did not try and interfere with endosomal actin. In my eyes this is still a weakness in the analysis of the somatic actin puncta and should be rectified.

2nd Revision - authors' response

6 September 2019

Referee #2:

In this revised version, Meka et al. have addressed the points made in my review, in particular the experimental ones, with a clearer presentation of results. It's still not very clear to me from the manuscript that the authors favor a model where actin at the tip of neurites would have a negative effect on neurite growth and how experiments that examines neurite growth as an output fit in this model (as stated and explained in the response to reviewers), but this is a presentation issue that is quite minor.

We thank the Referee #2 for his/her comments. Since we could not convince the

referee, we now added a few lines in the first paragraph of the discussion part, leaving the issue as an open question.

Other very minor points: the second to last phrase of the introduction's first paragraph ("Thus, adding weight...") seems to be missing a verb; and in the last phrase of the same paragraph ("To test this possibility, we performed..."), I guess the authors mean "experiments" rather than "methods". I think the manuscript is now suitable for publication in EMBO Reports, and look forward to this interesting addition to the literature on actin organization in neurons.

We thank the Referee #2 for pointing out these two mistakes. We have changed the text accordingly.

We would like to thank this Referee for his/her insightful comments which helped us to improve our manuscript.

Referee #3:

The authors have now answered most of my queries and have improved the manuscript. It is a bit curious that the authors chose only to use Brefeldin A to look at endomembranes as possible source of the actin puncta and did not try and interfere with endosomal actin. In my eyes this is still a weakness in the analysis of the somatic actin puncta and should be rectified.

We agree with the Referee #3 on his/her concern, we now included a few lines in the second last paragraph of the Results part.

Lastly, we would like to thank this Referee for his/her helpful comments to improve our manuscript.

3rd Editorial Decision

27 September 2019

Thank you for submitting your revised manuscript. I have looked at everything and all looks fine. Therefore I am very pleased to accept your manuscript for publication in EMBO Reports.

Congratulations on a very nice study!

Corresponding Author Name: Durga Praveen Meka and Froylan Calderon de Anda

Manuscript Number: EMBOR-2019-47743-T